# GENDR: LIGHTEN GENERATIVE DETAIL RESTORATION

**Yan Wang, Shijie Zhao**[*]**, Kexin Zhang, Junlin Li, Li Zhang**
ByteDance
{wangyan.my, zhaoshijie.0526}@bytedance.com

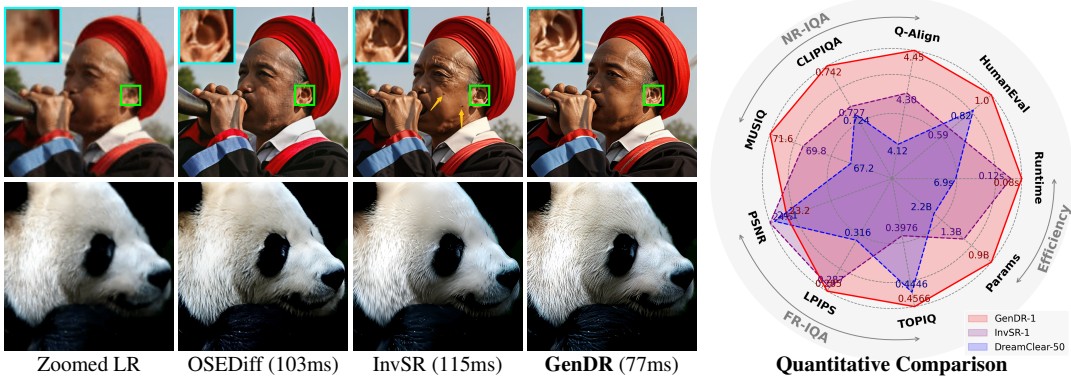

Figure 1: Visual comparison between proposed GenDR and recent state-of-the-art diffusion-based models (our method provides more details and higher fidelity with fewer time cost tested with $512^2$px on A100) (left) and quantitative comparison of representative SR methods (right), both of which demonstrate the advanced performance of the proposed GenDR. (Zoom in for best view.)

## ABSTRACT

Although recent research applying text-to-image (T2I) diffusion models to real-world super-resolution (SR) has achieved remarkable progress, the misalignment of their targets leads to a suboptimal trade-off between inference speed and detail fidelity. Specifically, the T2I task requires multiple inference steps to synthesize images matching to prompts and reduces the latent dimension to lower generating difficulty. Contrariwise, SR can restore high-frequency details in fewer inference steps, but it necessitates a more reliable variational auto-encoder (VAE) to preserve input information. However, most diffusion-based SRs are multistep and use 4-channel VAEs, while existing models with 16-channel VAEs are overqualified diffusion transformers, *e.g.*, FLUX (12B). To align the target, we present a one-step diffusion model for generative detail restoration, GenDR, distilled from a tailored diffusion model with a larger latent space. In detail, we train a new SD2.1-VAE16 (0.9B) via representation alignment to expand the latent space without increasing the model size. Regarding step distillation, we propose consistent score identity distillation (CiD) that incorporates SR task-specific loss into score distillation to leverage more SR priors and align the training target. Furthermore, we extend CiD with adversarial learning and representation alignment (CiDA) to enhance perceptual quality and accelerate training. We also polish the pipeline to achieve a more efficient inference. Experimental results demonstrate that GenDR achieves state-of-the-art performance in both quantitative metrics and visual fidelity.

## 1 INTRODUCTION

Image super-resolution (SR) is a classical low-level problem to recover a high-resolution (HR) image from the low-resolution (LR) version (Dong et al., 2016; Lim et al., 2017). Its core aim is to repair missing high-frequency information from complex or unknown degradation in real-world scenarios with the help of learned priors. To reconstruct more realistically, the existing

---

[*]Corresponding Author.

methods SRGAN,RealESRGAN introduce generative adversarial networks (GAN) to reproduce the details and have revealed advanced performance. Recently, modernized text-to-image (T2I) models, *e.g.*, *Stable Diffusion* (SD) and *PixArt*, have demonstrated their ability to synthesize high-resolution images with photographic quality, offering a new paradigm to replenish details.

To assign an image SR objective to these T2I-oriented diffusion models, Wang et al. (2024b); Lin et al. (2024) leverage additional control modules such as controlnet and encoder to guide a T2I model to generate HR images from the noise. Although these methods exhibit superior restoration quality compared to GAN-based methods, two disturbing problems persist that restrict the practical usage of diffusion-based SR: *slow inference speed* and *inferior detail fidelity*. To address these issues, OSEDiff (Wu et al., 2024b) employs variational score distillation (VSD) (Wang et al., 2024d) through low-rank adaptation (LoRA) (Hu et al., 2022) to directly distill a one-step SR model from the T2I model, which significantly improves throughputs but weakens generation quality. Dream-Clear (Ai et al., 2025) employs two controlnets and MLLM-generated prompts to guide PixArt-$\alpha$ (Chen et al., 2023a), generating more faithful results. However, existing methods are caught in the dilemma that improving detail fidelity brings computational overhead (larger base model/additional assistant module), leading to inefficiency while accelerating inference results in unacceptable performance drop since they neglect critical underlying.

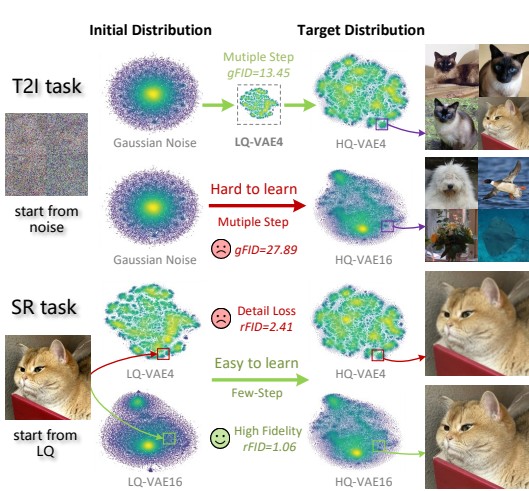

Figure 2: **Motivation**: divergent task objectives make dilemma. Compared to T2I task, SR task generates only details from LQ to HQ but needs to preserve more information from LQ, demanding fewer inference steps and larger latent capability. *We visualize the latent distribution on ImageNet-val ImageNet with t-SNE.*

As shown in Fig. 2, we identify divergent task objectives between T2I *requiring more inference steps and low-dimensional latent space* and SR *requiring fewer inference steps and high-dimensional latent space* as the main contributors to the above problems. This finding is based on the following observations:

- *Divergent generating difficulty*: T2I models start from a sampled Gaussian noise to generate both low-frequency content and high-frequency details aligned with textual prompts. Nevertheless, the low-frequency structural and semantic components are well-defined in LR for the SR task, primarily demanding detail generation. Consequently, an ideal SR-tailored diffusion model can theoretically use fewer inference steps since it focuses on details and has no need to align with text.

- *Divergent reconstruction demands*: For T2I, expanding latent space inherently increases model complexity and training difficulty, leading most frameworks to adopt a 4-channel VAE to balance synthesis quality and computational efficiency. Although this trade-off optimizes performance for T2I, it becomes a barrier for reconstruction tasks like SR. VAE4s stumble to preserve intricate details within compressed latent representation, resulting in irreversible detail loss and structural distortion (Dai et al., 2023; Esser et al., 2024).

To address these limitations holistically, this work provides a systematic solution, namely GenDR, for real-world super-resolution (SR), particularly targeting faithful and intricate Generative Detail Restoration. To build this model, we introduce several innovations:

- *Tailored Base Model*. Since SR tasks demand a larger latent space, we construct GenDR with a 16-channel base model. However, existing models (DiTs like SD3.5 and FLUX) are overqualified for generating details and prohibitively large to exacerbate the dilemma between quality and speed. For instance, to execute ×4 SR for 256×256 input on a FLUX-based SR, the one-step processing costs over 40GB GPU memory and 1.4s runtime, which is 5.3× and 11.4× larger than SD2.1-based

models. Thus, based on SD2.1 and an open-source VAE16, we construct an SD2.1-VAE16 as a suitable base model for diffusion-based restoration tasks.

- *Advanced Step Distillation*. To minimize the inference process to one step, we perform step distillation for GenDR. Unlike existing methods (Wu et al., 2024b; Sauer et al., 2024b) that directly employ score distillation from T2I, we integrate task-specific consistent priors from SR into score identity distillation (SiD) (Zhou et al., 2024b), proposing Consistent score identity Distillation (CiD), which mitigates adverse effects caused by inconsistencies in the training distribution and over-reliance on imperfect score functions. Furthermore, we introduce CiDA, which incorporates CiD with representation Alignment and Adversarial learning to accelerate training and restore high-diversity details, avoiding the "fakeness" of AI-generated images. In practice, we apply low-rank adaptation (LoRA) and model-sharing strategies to implement the CiDA training scheme more efficiently.

- *Simplified Pipeline*. We construct a simplified diffusion pipeline comprised solely of VAE and UNet. We remove the scheduler and conditioning modules and use fixed textual embeddings to enable efficient deployment.

Overall, GenDR gains remarkable improvement over existing models in objective quality/efficiency metrics (Tab. 1), subjective visual comparison (Fig. 5), and human evaluation (Fig. 6).

## 2 RELATED WORK

### 2.1 GENERATIVE PRIOR FOR SR

Existing generative prior-based SR methods can be broadly categorized into GAN prior-based and diffusion prior-based paradigms. In detail, **GAN** can produce realistic synthesis by enforcing an adversarial updating between the discriminator and generator. Given the generator $\mathcal{G}$ and discriminator $\mathcal{D}$, GAN cyclically optimizes them with varied adversarial losses:

$$
\begin{aligned}
\mathcal{L}^{\mathcal{D}} &= \mathbb{E}_{\mathbf{x}_h, \mathbf{x}_g = \mathcal{G}(\mathbf{x}_l)} \left[ \ln(1 - \mathcal{D}(\mathbf{x}_g)) + \ln \mathcal{D}(\mathbf{x}_h) \right], \\
\mathcal{L}^{\mathcal{G}} &= \mathbb{E}_{\mathbf{x}_g} \ln \mathcal{D}(\mathbf{x}_g),
\end{aligned}
\tag{1}
$$

where $\mathbf{x}_g$, $\mathbf{x}_h$, and $\mathbf{x}_l$ are SR images, high-quality, and low-quality images for SR tasks. Based on the target, GAN-based SR models (Zhang et al., 2021; Wang et al., 2018) exploit adversarial training to synthesize visually pleasing details at the cost of occasional instability during optimization. As a milestone, Real-ESRGAN (Wang et al., 2021) introduced a high-order degradation datapipe to synthesize LR-HR training pairs.

**Diffusion model** decouples the image synthesis into forward and reverse processes in the latent space to stabilize the training and generated quality. For forward diffusion processing, the Gaussian noise $\epsilon \sim \mathcal{N}(0, \mathbf{I})$ is added with the time $t$-related variance $\beta_t \in (0, 1)$ to obtain the immediate latent $\mathbf{z} : \mathbf{z}_t = \bar{\alpha}_t \mathbf{z} + \bar{\beta}_t \epsilon$, where $\alpha_t = 1 - \beta_t$ and $\bar{\alpha}_t = \mathrm{sqrt}(\prod_{s=1}^{t} \alpha_s)$. During the reverse process, the diffusion model predicts the noise $\hat{\epsilon}$, thus obtaining the initial latent $\hat{\mathbf{z}}_0 = (\mathbf{z}_t - \bar{\beta}_t \hat{\epsilon})/\bar{\alpha}_t$. For SR tasks, a better-starting node, *i.e.*, the LR image $\mathbf{z}_l$, can enable an efficient and simplified reverse function to calculate restored latent $\mathbf{z}_g$:

$$
\mathbf{z}_g = \frac{\mathbf{z}_l - \bar{\beta}_{t_s} \epsilon_\theta(\mathbf{z}_l; t_s)}{\bar{\alpha}_{t_s}},
\tag{2}
$$

where $\epsilon_\theta$ represents the noise-predicting network. Diffusion-based SR models (Wu et al., 2024c; Zhang et al., 2024; Ai et al., 2025; Yang et al., 2024) finetune base model (*e.g.*, UNet and DiT) or train an extra conditioning model (*e.g.*, Encoder and ControlNet) to guide iterative generation that aligns to the low-quality input. For instance, DiffBIR (Lin et al., 2024) introduced a preclear module and IRControlNet to balance fidelity and detail generation. FaithDiff (Chen et al., 2024b) fine-tuned an auxiliary encoder for SDXL (Rombach et al., 2022). Recently, Yi et al. (2025) proposed Transfer VAE Training (TVT), which tried to solve fidelity problem by training better VAE for diffusion pipeline, which achieves the same compression rate as GenDR.

## 2.2 STEP DISTILLATION

To reduce inference cost, massive research has focused on distilling multiple-step diffusion processes into few-step frameworks. As a pioneering work, DreamFusion (Poole et al., 2022) introduced score distillation sampling (SDS) to transfer knowledge from diffusion to arbitrary generators, which paves the way for step distillation. ProlificDreamer (Wang et al., 2024d) and Diff-instruct (Luo et al., 2023) utilized variational score distillation to enhance generation diversity and stability through probabilistic refinement. SiD (Zhou et al., 2024b) further ameliorated the score distillation with identity transformation for stable training.

## 3 METHODOLOGY

### 3.1 SD2.1-VAE16: 16-CHANNELS VAE MATTERS

To tailor a base model with a larger latent space for the SR task, we develop the UNet from SD2.1[1] in the latent space of 16-channel VAE[2]. To speed up training, we conduct full-parameter optimization using the representation alignment strategy (Yu et al., 2024b). Given input image $\mathbf{x}_h$, intermediate latent $\mathbf{z}_t$, encoded output $\mathbf{h}_t = f_\theta(\mathbf{z}_t)$, REPA aligns projected results $\mathbf{h}_t$ with the representation $\mathbf{h}_\mathcal{E} = \mathcal{E}(\mathbf{x}_h)$ obtained by pre-trained encoder $\mathcal{E}$, $e.g.$ DINOv2 (Oquab et al., 2023):

$$\mathcal{L}^{(\text{repa})} = -\mathbb{E}_{\mathbf{x}_h,t} \left[ \frac{1}{N} \sum_{n=1}^{N} \text{sim} \left( \mathbf{h}_\mathcal{E}[n], h(\mathbf{h}_t[n]) \right) \right]. \tag{3}$$

In practice, we insert the multilayer perceptron (MLP) as $h$ after the first downsampling block in UNet. To further improve the model's comprehension of image quality attributes, $i.e.$, quality-assessment terminologies like *compressed* and *blurred*, we refine the base model following the DreamClear (Ai et al., 2025).

### 3.2 CID: CONSISTENT SCORE IDENTITY DISTILLATION

The existing score distillation methods (Poole et al., 2022; Wang et al., 2024d; Zhou et al., 2024b) are based on the insight: *If generator $\mathcal{G}_\theta$ produces outputs $\mathbb{P}_g$ approximating to real data distribution $\mathbb{P}_r$, the score model $\psi$ trained by $\mathbb{P}_g$ should converge to pretrained model $\phi$ trained by $\mathbb{P}_r$.*

**Score distillation**. Given the fixed real score model $\phi$ and updated fake score model $\psi$, the restoration network $\mathcal{G}_\theta$, generated latent $\mathbf{z}_g = \mathcal{G}_\theta(\mathbf{z}_l)$, sampled timestep $t$, and noise $\epsilon$, the VSD (Wang et al., 2024d) is to cyclingly update $\theta$ and $\psi$ by:

$$\nabla_\theta \mathcal{L}_\theta^{(\text{vsd})} = \mathbb{E}_{t,\epsilon,c,\mathbf{z}_t=\bar{\alpha}_t\mathbf{z}_g+\bar{\beta}_t\epsilon} \left[ \omega(t)(\epsilon_\phi(\mathbf{z}_t;t,c) - \epsilon_\psi(\mathbf{z}_t;t,c))\frac{\partial \mathbf{z}_g}{\partial \theta} \right],$$

$$\mathcal{L}_\psi = \mathbb{E}_{t,\epsilon,c,\mathbf{z}_t} \left[ ||\epsilon_\psi(\mathbf{z}_t;t,c) - \epsilon||^2 \right], \tag{4}$$

where $\omega(t)$ is a time-aware weighting function. $\epsilon_{\phi,\psi}$ represents scores (noise) predicted by real/fake score networks. $c$ indicates the prompt condition.

**Score identity distillation**. While VSD provides a tractable framework, it neglects the gradient of the fake score network and is difficult to ensure that $\epsilon_\psi$ achieves convergence, leading to unstable training and susceptibility to local optima due to inadequate convergence guarantees. Following Zhou et al. (2024b), we leverage score identity transformation to pre-estimate an ideal value $\epsilon_\psi(\mathbf{z}_t;t,c) = -\bar{\beta}_t\nabla_{\mathbf{z}_t}\log p_\theta(\mathbf{z}_t)$ for $\epsilon_\psi$, incidentally reducing the dependence on $\epsilon_\psi$:

$$\mathcal{L}_\theta^{(1)} = \mathbb{E}_{t,\epsilon,c,\mathbf{z}_t=\bar{\alpha}_t\mathbf{z}_g+\bar{\beta}_t\epsilon} \left[ ||\epsilon_\phi(\mathbf{z}_t;t,c) - \epsilon_\psi(\mathbf{z}_t;t,c))||^2 \right] \xrightarrow{\text{SiD}}$$

$$\mathcal{L}_\theta^{(2)} = \mathbb{E}_{t,\epsilon,c,\mathbf{z}_t} \left[ \omega(t)\langle \epsilon_\phi(\mathbf{z}_t;t,c) - \epsilon_\psi(\mathbf{z}_t;t,c), \epsilon_\phi(\mathbf{z}_t;t,c) - \epsilon \rangle \right]. \tag{5}$$

**Consistent score identity distillation**. Despite effectiveness, the above-mentioned distillation strategies are tailored for the T2I task. The existing SR diffusion models, $e.g.$, ADDSR (Xie

---

[1]huggingface: stabilityai/stable-diffusion-2-1

[2]huggingface: ostris/vae-kl-f8-d16

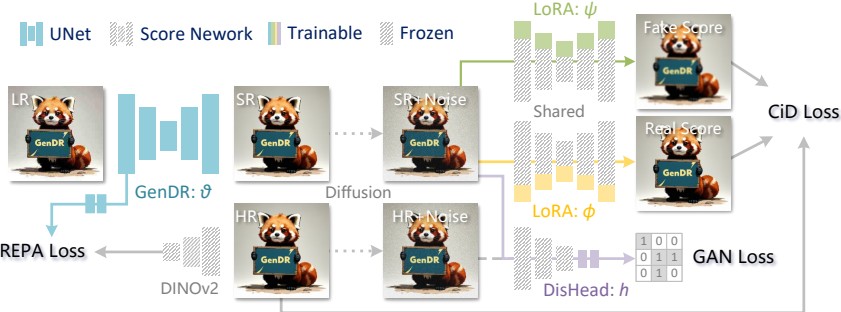

Figure 3: Illustration of the proposed CiDA training scheme for GenDR. GenDR and base score network are initialized with SD2.1-VAE16. The real and fake score networks are implemented by LoRA. LR latent is fed into GenDR to restore SR-latent and representation for REPA loss Eq. (3).

et al., 2024) and OSEDiff (Wu et al., 2024b) apply Eq. (4) directly Eq. (4) as an additional loss of regularization without any specific modification. However, for the SR task, directly using Eqs. (4) and (5) poses difficulties since *T2I (aligning text embedding)* and *SR (aligning image embedding) have distinct targets and varying training distributions, leading to quality and content inconsistency.* Consequently, these methods employ regression loss (L1, MSE) to ensure consistency between generated latent $\mathbf{z}_g$ and HR target $\mathbf{z}_h$:

$$\mathcal{L}_\theta^{(\mathrm{mse})} = \mathbb{E}_{t,c}\left[||\mathbf{z}_g - \mathbf{z}_h||^2\right]. \tag{6}$$

To address the issue, we optimize the "fixed" real score network $\phi$ with $z_h$ to align its output distribution with the target high-fidelity image manifold, which ensures the real score network generates reliable priors for distillation. Following Zhou et al. (2024a), we regulate Eqs. (4) and (5) in latent space and introduce classifier-free guidance (CFG) (Ho & Salimans, 2022) with quality-related prompts to enhance guidance quality. Overall, the primary CiD can be formulated as follows:

$$\begin{aligned}
\mathcal{L}_\psi &= \mathbb{E}_{t,\epsilon,c,\mathbf{z}_t=\bar{\alpha}_t\mathbf{z}_g+\bar{\beta}_t\epsilon}\left[||f_\psi(\mathbf{z}_t;t,c) - \mathbf{z}_g||^2\right], \\
\mathcal{L}_\phi &= \mathbb{E}_{t,\epsilon,c,z_t=\bar{\alpha}_t\mathbf{z}_h+\bar{\beta}_t\epsilon}\left[||f_\phi(\mathbf{z}_t;t,c) - \mathbf{z}_h||^2\right], \\
\tilde{\mathcal{L}}_\theta^{(2)} &= \mathbb{E}_{t,\epsilon,c,\mathbf{z}_t}\left[\omega(t)\langle f_{\phi,\kappa}(\mathbf{z}_t;t,c) - f_{\psi,\kappa}(\mathbf{z}_t;t,c), f_{\phi,\kappa}(\mathbf{z}_t;t,c) - \mathbf{z}_g\rangle\right],
\end{aligned} \tag{7}$$

where $f_{\phi,\psi}$ denotes scores (latent) predicted by real/fake score networks. $f_\kappa(\mathbf{z}_t;t,c) = f(\mathbf{z}_t;t,\varnothing) + \kappa \cdot [f(\mathbf{z}_t;t,c) - f(\mathbf{z}_t;t,\varnothing)]$, $\varnothing$ represents empty prompt embeddings.

Similar to VSD and SiD, the above step distillation framework is data-free and relies on generated results $\mathbf{z}_g$, which introduce instability due to $\mathbf{z}_g$'s fluctuating quality and content. To mitigate this, we leverage the optimal latent for $\theta$ in Eq. (6) to replace $\mathbf{z}_g$ with $\mathbf{z}_h$ as an identity transformation.

$$\mathcal{L}_\theta^{(3)} = \mathbb{E}_{t,\epsilon,c,\mathbf{z}_t}\left[\omega(t)\langle f_{\phi,\kappa}(\mathbf{z}_t;t,c) - f_{\psi,\kappa}(\mathbf{z}_t;t,c), f_{\phi,\kappa}(\mathbf{z}_t;t,c) - \mathbf{z}_h\rangle\right]. \tag{8}$$

Overall, to circumvent the inability to compute $\nabla_\theta\psi$, we follow Zhou et al. (2024b); Huang et al. (2024) to combine $\mathcal{L}_\theta^{(1)}$ and $\mathcal{L}_\theta^{(3)}$ through empirical weighting $\xi$ to ensure unbiased optimization:

$$\mathcal{L}_\theta^{(\mathrm{cid})} = \mathcal{L}_\theta^{(3)} - \xi\mathcal{L}_\theta^{(1)}. \tag{9}$$

### 3.3 CiDA: CiD with Adversary and Alignment

Following DMD2 (Yin et al., 2024) and REPA (Yu et al., 2024b), we further develop the CiD (Eq. (9)) with adversarial learning and representation alignment. Based on Eq. (1), we examine the generated latent $\mathbf{z}_g$ with pre-trained unet $\phi$ and extra discriminative heads $h$:

$$\mathcal{L}_\theta^{(\mathrm{adv})} = \frac{1}{H'W'}\sum_{i=1}^{H'}\sum_{j=1}^{W'}\ln h(f_\phi(\mathbf{z}_g))[i,j], \tag{10}$$

where $H' \times W'$ denotes the spatial dimensions of the discriminator's output feature map.

We also reintroduce representation alignment in Eq. (3) as an effective regularization term. Combined with the CiD objective in Eq. (9) and adversarial loss in Eq. (10), the final target loss integrates:

$$\mathcal{L}_\theta^{\text{(cida)}} = \lambda_1 \mathcal{L}_\theta^{\text{(cid)}} + \lambda_2 \mathcal{L}_\theta^{\text{(adv)}} + \lambda_3 \mathcal{L}_\theta^{\text{(repa)}}, \qquad (11)$$

where $\lambda_{1/2/3}$ are weighting coefficients balancing the contributions of distillation, adversarial, and alignment terms.

**Implemetation with LoRA**. CiDA consists of one trainable generator and two trainable score networks, which cost huge GPU memory and computation resources for parameter updating across three UNets. To alleviate this burden, we utilize the low-rank adaptation (LoRA) Hu et al. (2022) for fake/real score networks to reduce the optimizable parameters. In addition, we share the base model for score networks and the feature extractor of the discriminator to further reduce the memory footprint of storing local models. In Fig. 3, we illustrate the detailed implementation of the proposed CiDA framework. In Appendix A.2, we present the detailed training schema for CiDA.

### 3.4 SIMPLIFIED DIFFUSION PIPELINE

To enable efficient inference without relying on complex preclear models or conditioning models, we design GenDR using a simplified architecture comprising only a UNet and VAE. Since GenDR executes a one-step calculation, we eliminate the scheduler by empirically assigning $\bar{\alpha}_t = \bar{\beta}_t = 0.5$ across all timesteps $t$. Additionally, we discard the text encoder and tokenizer, replacing them with several fixed-prompt embeddings stored locally to reduce computational overhead and ensure deterministic generation. Fig. 4 shows the extremely laconic pipeline of GenDR.

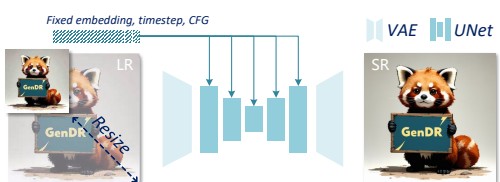

Figure 4: Illustration of proposed GenDR pipeline.

## 4 EXPERIMENTS

### 4.1 EXPERIMENTAL SETUP

**Training details**. For SD2.1-VAE16, we train it on selected high-quality images (Wang et al., 2022; Rombach et al., 2022). The training resolution is $512\times512$ and $1024\times1024$, which aligns to SD2.1-base and SDXL. We use ZeRO2-Offload (Rajbhandari et al., 2020) and gradient accumulation (steps=8) to extend the batch size to 2048 on 8 NVIDIA A100 GPUs. The fixed learning rate $1\text{e}^{-5}$, and default AdamW is adopted to optimize UNet during 100k iterations.

For GenDR, we initialize $\mathcal{G}_\theta$ and $f_{\phi,\psi}$ with SD2.1-VAE16-512px and conduct full parameter training for $\mathcal{G}_\theta$ on the LSDIR (Li et al., 2023), FFHQ (Karras et al., 2019), and above-selected images. As to $f_{\phi,\psi}$, we set LoRA rank and alpha as 64 and 128, respectively. Referring to previous setups (Lin et al., 2024; Wu et al., 2024c;b), we randomly crop $512\times512$ image patches to generate LR-HR pairs through a mixed pipeline of Real-ESRGAN (Wang et al., 2021) and APISR (Wang et al., 2024a). During 50k iteration training, networks are optimized via AdamW optimizers with batchsize 1024 and learning rate $1\text{e}^{-5}$ on 8 NVIDIA A100-80G. We set start-timestep $t_s = 500$ for GenDR, and randomly sample timesteps $t \in \{20, \ldots, 979\}$ for CiDA score networks $f_{\phi,\psi}$. The loss coefficients $\lambda_{1,2,3}$ are 10 (first 10k)/1, 0.01, and 0.1, respectively. Specifically, we use $\omega(t) = C/\text{sg}[||\mathbf{z}_h - f_{\phi,\kappa}(\mathbf{z}_t; t, c)||]$ as the time-related function to modulate CiDA loss.

**Testing details**. Following recent work (Yue et al., 2024b), we evaluate the proposed GenDR with the synthetic dataset ImageNet-Test (Deng et al., 2009) and several real-world test sets, RealSR (Cai et al., 2019), RealSet80 (Yue et al., 2024b), and RealLR200 (Wu et al., 2024c). The ImageNet-Test contains 3000 images under multiple complicated degradations. RealSR has 100 HR-LR image pairs taken by Nikon and Canon cameras. The RealSet80 and RealLR200 contain 80 and 200 low-resolution images without ground truth. In the test phase, we use official codes and settings to examine the

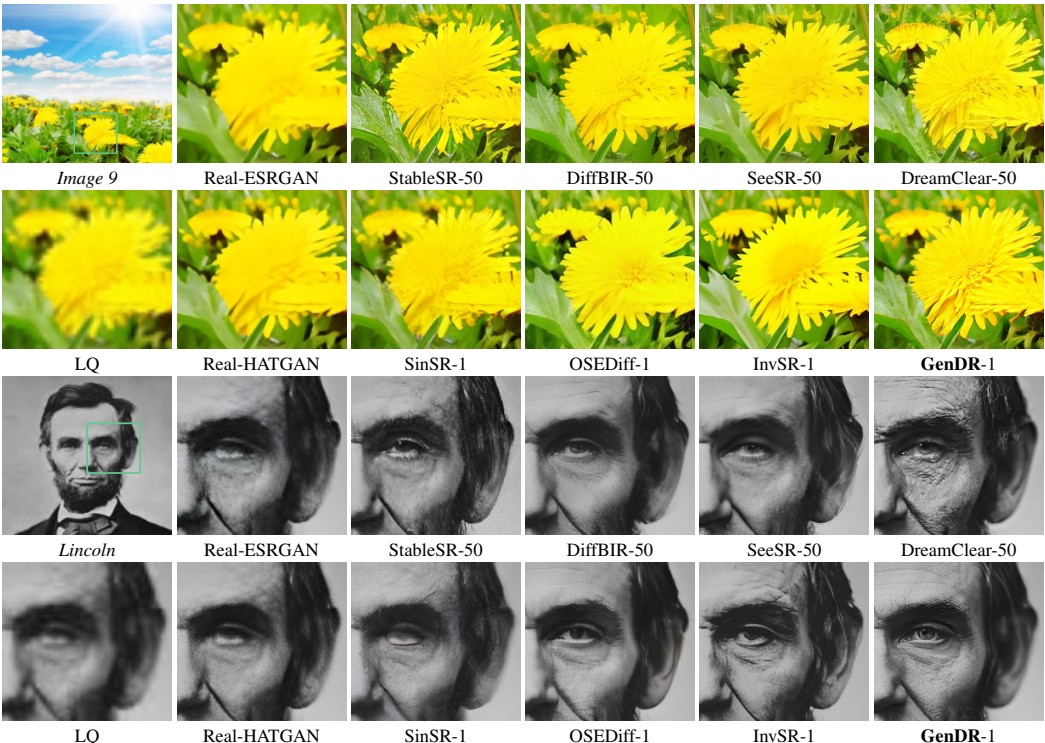

Figure 5: Visual comparison of GenDR with other methods for ×4 task on RealSet80 dataset.

compared models for fairness. For GenDR, the positive/negative prompts are fixed to provide a general discription[3].

**Evaluation metrics and compared methods**. We leverage seven commonly used metrics to evaluate image quality, including full-reference metrics (FR-IQA): PSNR, SSIM (Wang et al., 2004), LPIPS (Zhang et al., 2018), and TOPIQ (Chen et al., 2024a), and no-reference metrics (NR-IQA): PI (Blau et al., 2018), NIQE (Mittal et al., 2012), LIQE (Zhang et al., 2023), MANIQA (Yang et al., 2022), ClipIQA (Wang et al., 2023), MUSIQ (Ke et al., 2021), ARNIQA (Agnolucci et al., 2024), Q-Align (Quality) (Wu et al., 2024a), and DeQA (You et al., 2025). We also conduct user and MLLM preference studies to obtain a comprehensive evaluation of image quality.

## 4.2 COMPARISON WITH STATE-OF-THE-ARTS

**Quantitative comparison**. To comprehensively evaluate the performance of GenDR, we compare it with numerous state-of-the-art methods, including GAN-based models: RealESRAN (Wang et al., 2021) and Real-HATGAN (Chen et al., 2023b), multiple-step diffusion models: StableSR (Wang et al., 2024b), DiffBIR Lin et al. (2024), SeeSR (Wu et al., 2024c), and DreamClear (Ai et al., 2025), one-step diffusion models: SinSR (Wang et al., 2024c), OSEDiff (Wu et al., 2024b), and InvSR (Yue et al., 2024a). Inclusively, our GenDR obtains superior restoration quality across all three benchmark datasets. As shown in Tab. 1, GenDR surpasses all existing one-step models by a large margin and can compete with multiple-step diffusion models. Specifically, GenDR achieves the highest LIQE, MUSIQ, and Q-Align on all benchmarks. Moreover, we add total parameters and inference time in Tab. 1, where the GenDR is the fastest and second smallest diffusion model. Compared to recent DreamClear, the GenDR gains about 89.5× acceleration and only uses half of the parameters.

**Qualitative comparision**. Fig. 5 exhibits the visual comparison for GenDR with other approaches. Generally, GenDR presents the best quality in terms of blurry removal and detail recovery. For *Image*

---

[3]*"realism photo, best quality, realistic detailed, clean, high-resolution, best quality, smooth plain area, high-fidelity, clear edge, clean details without messy patterns, high-resolution, no noise, high-fidelity, 4K, 8K, perfect without deformations, photo taken in the style of Canon EOS −style raw"*

Table 1: Quantitative comparison (average Parameters, Inference time, and IQA metrics) on both synthetic and real-world benchmarks. The sampling step number is marked in the format of "Method name-Steps" for diffusion-based methods. The best results for all methods are highlighted in **bold** and underlined, while the best one-step diffusion methods are reported in *italic*.

| | Methods | #Params↓ | Metrics | | | | | | | |
| --- | --- | --- | --- | --- | --- | --- | --- | --- | --- | --- |
| | | | PSNR↑ | SSIM↑ | LPIPS↓ | NIQE↓ | LIQE↑ | ClipIQA↑ | MUSIQ↑ | Q-Align↑ |
| *ImageNet-Test* | Real-ESRGAN | GAN-CNN | 26.62 | 0.7523 | 0.2303 | 4.4909 | 3.8414 | 0.5090 | 64.81 | 3.4230 |
| | Real-HATGAN | GAN-Trans | **27.15** | **0.7690** | **0.2044** | 4.7834 | 3.5717 | 0.4594 | 63.43 | 3.3244 |
| | StableSR-50 | SD 2.1-base | 26.00 | 0.7317 | 0.2327 | 4.9378 | 3.6187 | 0.5768 | 64.54 | 3.4378 |
| | DiffBIR-50 | SD 2.1-base | 25.45 | 0.6651 | 0.2876 | 4.9289 | 4.6378 | 0.7486 | 73.04 | 4.3228 |
| | SeeSR-50 | SD 2-base | 25.73 | 0.7072 | 0.2467 | 4.3530 | 4.5384 | 0.6981 | 72.25 | 4.2412 |
| | DreamClear-50 | PixArt-$\alpha$ | 24.76 | 0.6672 | 0.2463 | 5.3787 | 4.4298 | **0.7646** | 70.08 | 4.0919 |
| | SinSR-1 | LDM | *26.98* | *0.7308* | *0.2288* | 5.2506 | 3.9410 | 0.6607 | 67.70 | 3.8090 |
| | OSEDiff-1 | SD 2.1-base | 24.82 | 0.7017 | 0.2431 | *4.2786* | *4.5609* | 0.6778 | 71.74 | 4.0674 |
| | InvSR-1 | SD Turbo | 23.81 | 0.6777 | 0.2547 | 4.3935 | 4.5601 | 0.7114 | 72.38 | 3.9867 |
| | **GenDR**-1 | SD 2.1-VAE16 | 24.14 | 0.6878 | 0.2652 | ***4.1336*** | ***4.8096*** | *0.7395* | ***74.68*** | ***4.3612*** |
| *RealISR* | Real-ESRGAN | 16.70M | 25.85 | 0.7734 | 0.2729 | 4.6788 | 3.3372 | 0.4901 | 59.69 | 3.9185 |
| | Real-HATGAN | 20.77M | 26.22 | **0.7894** | **0.2409** | 5.1189 | 2.8875 | 0.4336 | 58.41 | 3.8353 |
| | StableSR-50 | 1410M | 24.52 | 0.6733 | 0.3658 | **3.4665** | 3.5612 | 0.6897 | 66.87 | 3.9862 |
| | DiffBIR-50 | 1717M | **26.28** | 0.7251 | 0.3187 | 5.8009 | 3.3588 | 0.6743 | 64.28 | 3.9182 |
| | SeeSR-50 | 2524M | 26.19 | 0.7555 | 0.2809 | 4.5366 | 3.7728 | 0.6826 | 66.31 | 3.9862 |
| | DreamClear-50 | 2212M | 24.14 | 0.6963 | 0.3155 | 3.9661 | 3.5452 | 0.6730 | 63.74 | 3.9705 |
| | SinSR-1 | 119M | *25.99* | 0.7072 | 0.4022 | 6.2412 | 3.0034 | 0.6670 | 59.22 | 3.8800 |
| | OSEDiff-1 | 1775M | 24.57 | 0.7202 | 0.3036 | 4.3408 | 3.9634 | 0.6829 | 67.30 | 4.0664 |
| | InvSR-1 | 1289M | 24.50 | *0.7262* | 0.2872 | 4.2218 | 4.0346 | 0.6919 | 67.47 | 4.2085 |
| | **GenDR**-1 | 933M | 23.18 | 0.7135 | *0.2859* | *4.1588* | ***4.1906*** | ***0.7014*** | ***68.36*** | ***4.2388*** |

| | Methods | Runtime↓ | Metrics | | | | | | | |
| --- | --- | --- | --- | --- | --- | --- | --- | --- | --- | --- |
| | | | PI↓ | ARNIQA↑ | DeQA↑ | NIQE↓ | LIQE↑ | ClipIQA↑ | MUSIQ↑ | Q-Align↑ |
| *RealSet80* | Real-ESRGAN | 36ms | 3.8843 | 0.6538 | 3.8906 | 4.1517 | 3.7392 | 0.6190 | 64.49 | 4.1696 |
| | Real-HATGAN | 116ms | 4.1817 | 0.6578 | 3.8444 | 4.4705 | 3.4927 | 0.5502 | 63.21 | 4.1077 |
| | StableSR-50 | 3731ms | **3.0314** | 0.6776 | 3.8787 | **3.3999** | 3.8516 | 0.7399 | 67.58 | 4.0870 |
| | DiffBIR-50 | 6213ms | 3.9544 | 0.6802 | 4.0668 | 5.1389 | 4.0472 | 0.7404 | 68.72 | 4.3206 |
| | SeeSR-50 | 6359ms | 3.7454 | 0.7170 | 4.0728 | 4.3749 | 4.2797 | 0.7124 | 69.74 | 4.3056 |
| | DreamClear-50 | 6892ms* | 3.4157 | 0.6738 | 3.9429 | 3.7257 | 3.9628 | 0.7242 | 67.22 | 4.1206 |
| | SinSR-1 | 120ms | 4.2697 | 0.6638 | 3.9713 | 5.6103 | 3.5957 | 0.6634 | 63.79 | 4.0954 |
| | OSEDiff-1 | 103ms | 3.6894 | 0.6883 | 4.0916 | 3.9763 | 4.1298 | 0.7037 | 69.19 | 4.3057 |
| | InvSR-1 | 115ms | *3.4524* | 0.7172 | 4.0045 | 4.0266 | 4.2906 | 0.7271 | 69.79 | 4.3014 |
| | **GenDR**-1 | 77ms | 3.5518 | ***0.7321*** | ***4.1289*** | *3.9750* | ***4.5248*** | ***0.7424*** | ***71.57*** | ***4.4525*** |

$\star$: inference time calculated under using 3 A100 GPU to run an image.

*9* from RealSet80, GenDR is the only model to restore the entire chrysanthemum and correctly split foreground and background. As to *Image Lincoln*, GenDR recovers clearer eyes and more natural skin from the degraded input. The visual results demonstrate the splendid recovery performance of the proposed GenDR. In Appendix A.4.4, we provide more visual comparisons.

**User studies and MLLM evaluation**. For a thorough comparison between diffusion-based SR, we conduct user studies to compare images restored by GenDR against SinSR, OSEDiff, and DreamClear. Fig. 6 shows that our GenDR convincingly outperforms other diffusion-based models. For further evaluation, we use MLLMs like Q-Align and DeQA, to enable human-like evaluation. Particularly, GenDR surpasses the second place by a large margin. Compared to recent InvSR, Q-Align and DeQA are improved by 0.12 and 0.15.

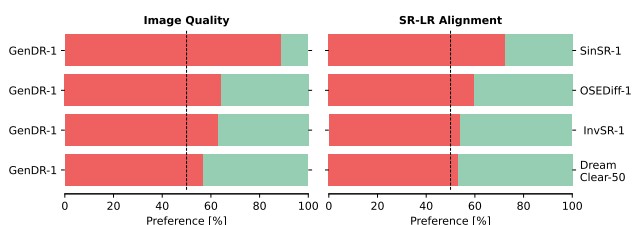

Figure 6: User studies on image quality and alignment.

Table 2: Ablation study on varied step distillation strategy for OSEDiff and GenDR on RealSet80.

| SD2.1 | Strategy | Metrics↑ | | | |
|---|---|---|---|---|---|
| | | LIQE | ClipIQA | MUSIQ | Q-Align |
| VAE4 | VSD | 4.1298 | 0.7037 | 69.19 | 4.3057 |
| | CiDA | 4.3184 | 0.7230 | 70.13 | 4.3858 |
| VAE16 | VSD | 4.1248 | 0.6911 | 68.82 | 4.3732 |
| | SiD | 4.2528 | 0.7016 | 69.33 | 4.3912 |
| | CiD | 4.4432 | 0.7150 | 70.61 | 4.4278 |
| | CiDA | **4.5248** | **0.7424** | **71.57** | **4.4525** |

Table 3: Ablation study on varied prompt generating setting for GenDR.

| Method | Prompt Extract | Prompt Embed | Time | #Params | #MACs | MUSIQ |
|---|---|---|---|---|---|---|
| Null | - | 15ms | 92ms | 1263M | 1637G | 70.66 |
| DAPE | 21ms | 15ms | 113ms | 1775M | 2638G | **71.74** |
| Qwen2.5 | 3.09s | 16ms | 3.18s | 8.3B | - | 71.08 |
| Fixed | - | 15ms | 92ms | 1263M | 1637G | 71.57 |
| **Fixed** | - | - | **77ms** | **933M** | **1623G** | 71.57 |

Table 4: Quantitative comparison between stable diffusions on 20k prompts from COCO 2014.

| Methods | GenEval↑ | | | | | | | FID↓ |
|---|---|---|---|---|---|---|---|---|
| | **Overall** | Single Obj. | Two Obj. | Counting | Colors | Position | Color Attr. | |
| SD1.5 | 0.43 | 0.97 | 0.38 | 0.38 | 0.76 | 0.04 | 0.06 | 13.45 |
| SD2.1 | 0.50 | 0.98 | 0.51 | 0.44 | **0.85** | 0.07 | 0.17 | **13.45** |
| **SD2.1-VAE16** | 0.48 (-0.02) | 0.98 | 0.53 | 0.35 | 0.79 | 0.08 | 0.12 | 27.89 (+14.44) |
| SDXL | **0.55** | **0.98** | **0.74** | 0.39 | **0.85** | **0.15** | **0.23** | 18.4 |

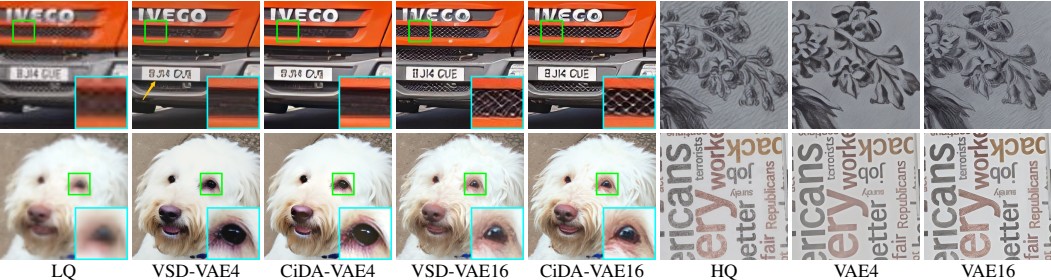

Figure 7: Visual comparison between VAE4 and VAE16, VSD and CiDA on RealSet80 and RealSR.

## 4.3 ABLATION STUDY

**Effectiveness of CiD and CiDA**. To examine the effectiveness of the proposed distillation approach, we use VSD (Wang et al., 2024d), SiD (Zhou et al., 2024b), CiD, and CiDA to distill SD2.1 (OSEDiff) and SD2.1-VAE16 (GenDR). Tab. 2 shows the quantitative comparisons, where the proposed CiDA yields a 0.08 improvement on Q-Align for both OSEDiff and GenDR. Specific to each loss function, the CiD accounts for 0.05 on Q-Align, and adversarial learning contributes the remaining 0.03. In Fig. 7, we provide a visual comparison for models trained by CiDA and VSD, where CiDA can provide vivid details and alleviate structural distortion.

**Effectiveness of 16 channel VAE**. Since GenDR is based on proposed SD2.1-VAE16, we first evaluate the new SD2.1-VAE16 in Tab. 4 by calculating GenEval (Ghosh et al., 2023) and FID on COCO 2014 (Lin et al., 2014), where the performance of SD2.1-VAE16 slightly decreases compared to SD2.1, indicating that in the T2I task, 4-channel VAE4 is more preferable than 16-channel for SD2.1. However, the performance reverses in the SR task. As exhibited in Tab. 2, compared to the 4-channel VAE model, SD2.1-VAE16 has similar or even higher objective scores. Moreover, the subjective comparison in Fig. 7 shows that VAE16 preserves more details and generates faithfully. Specifically, in the upper left case, the SD2.1-VAE16-based model restores the license plate and inlet grille. The results demonstrate that applying a 16-channel VAE matters for even small diffusion UNet in the SR task.

**Effectiveness of simplified pipeline**. For more efficient inference, we replace the text encoding modules and scheduler in GenDR with fixed embeddings. In Tab. 3, we compare the proposed GenDR pipeline with different prompt extractors (DAPE (Wu et al., 2024c), Qwen2.5VL-7B (Team, 2024)) and strategies, where our design is the most efficient and maintains a similar IQA score with content-related prompts. We provide more results on prompt selection in Appendix A.4.1.

## 5 LIMITATIONS

Despite our studies demonstrating that VAE-16 can alleviate detail fidelity degradation in the SR task, even for 0.9B UNet, further study on VAE with larger latent channels is not included due to the validated effectiveness of 16-channel VAE and the high training expenses for entire SD models. Another limitation lies in the training cost of CiDA. While we use LoRA and deepspeed to optimize the training process, CiDA needs large GPU memory, which restricts its extension in DiT models.

## 6 CONCLUSION

This work presents GenDR for real-world SR by tailoring the T2I model in an SR-favored manner, *i.e.*, using high-dimensional latent space to preserve and generate more details and one-step inference to improve efficiency. In detail, we develop an SD2.1-VAE16 providing a better trade-off between reconstruction quality and efficiency. Based on it, we introduce a consistent score identity distillation (CiD) that incorporates SR priors into T2I-oriented step diffusion to promote consistency and stability. Moreover, we improve CiD with adversarial learning and REPA to achieve better realistic details. Overall, GenDR obtains a remarkable quality-efficiency trade-off.

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

# A APPENDIX

## A.1 PROOF FOR CID

The key point for CiD is to replace $\mathbf{z}_g = f_{\theta^*}(\mathbf{z}_t, t)$ with a optimal $\mathbf{z}_h$ as the following identity transformation:

$$\mathbb{E}_{\mathbf{z}_h \sim \tilde{p}(\mathbf{z}_h), \mathbf{z}_l \sim p(\mathbf{z}_l)}[\langle f_\phi(\mathbf{z}_t, t), \mathbf{z}_g \rangle] = \mathbb{E}_{\mathbf{z}_h \sim \tilde{p}(\mathbf{z}_h), \mathbf{z}_l \sim p(\mathbf{z}_l)}[\langle f_\phi(\mathbf{z}_t, t), \mathbf{z}_h \rangle]. \tag{12}$$

MSE loss between SR and HR latent Eq. (6) can be formulated as:

$$\theta^* = \arg\min_\theta \mathbb{E}_{\mathbf{z}_h \sim \tilde{p}(\mathbf{z}_h), \mathbf{z}_l \sim p(\mathbf{z}_l)}[||f_\theta(\mathbf{z}_t, t) - \mathbf{z}_h||^2]. \tag{13}$$

Apparently, the optimal solution for Eq. (13) is:

$$f_{\theta^*}(\mathbf{z}_t, t) = \mathbb{E}_{\mathbf{z}_h \sim p(\mathbf{z}_h|\mathbf{z}_t)}[\mathbf{z}_h]. \tag{14}$$

Then, CiD is proved by replacing the optimal value for $\mathbf{z}_g$:

$$\begin{aligned}
&\mathbb{E}_{\mathbf{z}_h \sim \tilde{p}(\mathbf{z}_h), \mathbf{z}_l \sim p(\mathbf{z}_l)}[\langle f_\phi(\mathbf{z}_t, t), \mathbf{z}_g \rangle] \\
&= \mathbb{E}_{\mathbf{z}_t \sim p(\mathbf{z}_t)}[\langle f_\phi(\mathbf{z}_t, t), f_{\theta^*}(\mathbf{z}_t, t) \rangle] \\
&= \mathbb{E}_{\mathbf{z}_t \sim p(\mathbf{z}_t)}[\langle f_\phi(\mathbf{z}_t, t), \mathbb{E}_{\mathbf{z}_h \sim p(\mathbf{z}_h|\mathbf{z}_t)}[\mathbf{z}_h] \rangle] \\
&= \mathbb{E}_{\mathbf{z}_t \sim p(\mathbf{z}_t), \mathbf{z}_h \sim p(\mathbf{z}_h|\mathbf{z}_t)}[\langle f_\phi(\mathbf{z}_t, t), \mathbf{z}_h \rangle] \\
&= \mathbb{E}_{\mathbf{z}_h \sim \tilde{p}(\mathbf{z}_h), \mathbf{z}_l \sim p(\mathbf{z}_l)}[\langle f_\phi(\mathbf{z}_t, t), \mathbf{z}_h \rangle].
\end{aligned} \tag{15}$$

## A.2 ALOGRORITHM DETAILS

In Algorithms 1 and 2, we summarize the training schemes for SD2.1-VAE16 and GenDR, respectively. Specifically, we employ a two-stage training strategy (Yin et al., 2024; Zhou et al., 2024b) for GenDR to accelerate training. In the first stage, we train GenDR as a vanilla SR model with L1 loss and perceptual loss to align the output latent distribution with the target. This processing can effectively reduce the overall training time and improve the robustness of the training. In the second stage, we train GenDR with the proposed CiDA framework to make the results more realistic.

## A.3 EXPERIMENTAL DETAILS

### A.3.1 DATASET PREPARATION

The diffusion-based SR aims to train a generative model creating clear details, while the images from the existing open-sourced datasets are far from uniform in terms of quality, blur, blackness, aesthetics, and compression artifacts. Thus, we conduct a dataset preselection to filter images with IQA and aesthetics metrics, where 19.2% of images from the original datasets are preserved.

### A.3.2 USER AND MLLM PREFERENCE STUDIES

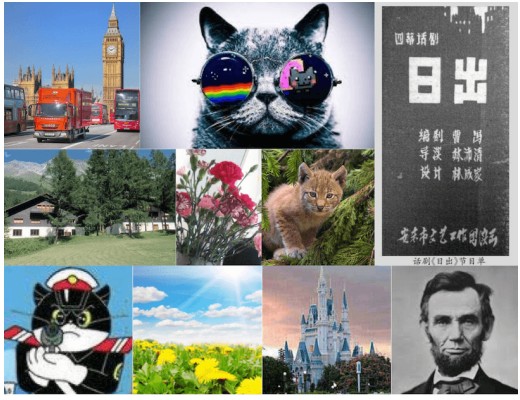

Figure 8: The LQ images used for the user study.

**User study**. Following Wu et al. (2024b); Sauer et al. (2024a), we conducted a user study to evaluate the human feedback of GenDR. We randomly sampled 10 real-world LQ images from RealSet80. During testing, LQ images and two SR images restored by GenDR and SinSR/InvSR/DreamClear were presented to volunteers to select the best-matched image in quality and fidelity attributes, respectively. In Fig. 8, we show the LQ images used for the user study.

**MLLM preference**. Since the user study compares only 10 images and 4 methods, we additionally develop an arena using DepictQA to make pairwise comparisons between the SR results of various models.

---

**Algorithm 1:** Training Scheme of SD2.1-VAE16

---

**Input:** Pretrained SD2.1 network $\epsilon_\mu$, SD2.1-VAE16 $\epsilon_\theta$, encoder of 16 channel VAE $\mathcal{E}_{\text{VAE}}$, pretrained DINOv2 encoder $\mathcal{E}_{\text{DINO}}$, $t_{\min} = 0$, $t_{\max} = 999$, $\lambda = 0.1$

**Initialization** $\theta \leftarrow \mu$

**repeat**

    Sample image $\mathbf{x}$ and prompt $c$ from batch $\mathcal{B}$; Calculate latent with $\mathbf{z} = \mathcal{E}_{\text{VAE}}(\mathbf{x})$ and extract representation $\mathbf{h}_\mathcal{E} = \mathcal{E}_{\text{VAE}}(\mathbf{x}_h)$,

    Sample $t \in \{t_{min}, \ldots, t_{\max}\}$ and $\epsilon_t \sim \mathcal{N}(0, \mathbf{I})$, calculate forward diffusion $\mathbf{z}_t = \bar{\alpha}_t \mathbf{z}_0 + \bar{\beta}_t \epsilon_t$, and let $(\epsilon_\theta, \mathbf{h}_t) = \epsilon_\theta(\mathbf{z}_t; t, c)$;

    Update $\theta$ with $\theta = \theta - \eta \nabla_\theta \mathcal{L}_\theta$, where

$$\mathcal{L}_\theta = ||\epsilon_\theta - \epsilon_t||_2^2 - \lambda \frac{\mathbf{h}_\mathcal{E} \cdot h(\mathbf{h}_t)}{||\mathbf{h}_\mathcal{E}|| \, ||h(\mathbf{h}_t)||}$$

**until** achieving convergence or the training budget is exhausted

**Output:** $\mathcal{G}_\theta$

---

**Algorithm 2:** Training Scheme of GenDR

---

**Input:** Pretrained diffusion network $f_\mu$, generator $\mathcal{G}_\theta$, real score network $f_\phi$, fake score network $f_\psi$, discriminative head $h_h$, encoder $\mathcal{E}$, $t_{\text{init}} = 499$, $t_{\min} = 20$, $t_{\max} = 979$, guidance scales $\kappa = 7.5$, $\lambda_{1,2,3} = 10, 0.01, 0.1$, $\xi = 1.2$

**Initialization** $\theta \leftarrow \mu$, initialize LoRA $\psi, \phi$, dishead $h$

▷ *Updating GenDR with vanilla SR loss*

**repeat**

    Sample latent pairs $(\mathbf{z}_h, \mathbf{z}_l)$ and prompt $c$ from batch $\mathcal{B}$; Sample $t \in \{t_{min}, \ldots, t_{\max}\}$ and $\epsilon_t \sim \mathcal{N}(0, \mathbf{I})$, and let $(\mathbf{z}_g, \mathbf{h}_t) = \mathcal{G}_\theta(\mathbf{z}_l; t, c)$

    Update $\mathcal{G}_\theta$ with $\theta = \theta - \eta \nabla_\theta \mathcal{L}_\theta$, where

$$\mathcal{L}_\theta^{(\text{reg})} = ||\mathbf{z}_g - \mathbf{z}_h||_2^2 + \mathcal{L}_\theta^{(\text{percep})}(\mathbf{z}_g, \mathbf{z}_h)$$

**until** achieving convergence or processing 20M images

▷ *Updating GenDR with CiDA*

**repeat**

    Sample latent pairs $(\mathbf{z}_h, \mathbf{z}_l)$ and prompt $c$ from batch $\mathcal{B}$; Sample $t \in \{t_{min}, \ldots, t_{\max}\}$ and $\epsilon_t \sim \mathcal{N}(0, \mathbf{I})$, and let $(\mathbf{z}_g, \mathbf{h}_t) = \mathcal{G}_\theta(\mathbf{z}_l; t, c)$; Stop gradient for $\mathbf{z}_g, \phi$; and let $\mathbf{z}_{g,t} = \bar{\alpha}_t \text{sg}[\mathbf{z}_g] + \bar{\beta}_t \epsilon_t$, $\mathbf{z}_{h,t} = \bar{\alpha}_t \mathbf{z}_h + \bar{\beta}_t \epsilon_t$

    Update $h$ with $h = h - \eta \nabla_h \mathcal{L}_h$, where

$$\mathcal{L}_h = \frac{1}{H'W'} \sum_{i=1}^{H'} \sum_{j=1}^{W'} \{\ln(1 - h_h(\hat{f}_{\text{sg}[\phi]}(\text{sg}[\mathbf{z}_g]))[i,j]) + \ln h_h(\hat{f}_{\text{sg}[\phi]}(\mathbf{z}_h))[i,j]\}$$

    Update $\phi$ with $\phi = \phi - \eta \nabla_\phi \mathcal{L}_\phi$ and $\psi$ with $\psi = \psi - \eta \nabla_\psi \mathcal{L}_\psi$, where

$$\mathcal{L}_\phi = ||f_\phi(\mathbf{z}_{h,t}; t, c) - \mathbf{z}_h||_2^2, \quad \mathcal{L}_\psi = ||f_\psi(\mathbf{z}_{g,t}; t, c) - \text{sg}[\mathbf{z}_g]||_2^2$$

    Sample $t \in \{t_{min}, \ldots, t_{\max}\}$ and $\epsilon_t \sim \mathcal{N}(0, \mathbf{I})$; Stop gradient for $\phi, \psi, h$, get $\omega(t) = C/\text{sg}[||\mathbf{z}_h - f_{\phi,\kappa}(\mathbf{z}_t; t, c)||]$, extract representation $\mathbf{h}_\mathcal{E} = \mathcal{E}(\mathbf{x}_h)$, and let $\mathbf{z}_t = \bar{\alpha}_t \mathbf{z}_g + \bar{\beta}_t \epsilon_t$

    Update $\mathcal{G}_\theta$ with $\theta = \theta - \eta \nabla_\theta \mathcal{L}_\theta$, where

$$\mathcal{L}_\theta^{(\text{cida})} = \lambda_1 \omega(t) \left(f_{\text{sg}[\phi],\kappa}(\mathbf{z}_t; t, c) - f_{\text{sg}[\psi],\kappa}(\mathbf{z}_t; t, c)\right)^T \left((1 - \xi)f_{\text{sg}[\phi],\kappa}(\mathbf{z}_t; t, c) - \mathbf{z}_h + \xi f_{\text{sg}[\psi],\kappa}(\mathbf{z}_t; t, c)\right)$$
$$+ \lambda_2 \frac{1}{H'W'} \sum_{i=1}^{H'} \sum_{j=1}^{W'} \ln h(f_{\text{sg}[\phi]}(\mathbf{z}_g))[i,j] - \lambda_3 \frac{\mathbf{h}_\mathcal{E} \cdot h(\mathbf{h}_t)}{||\mathbf{h}_\mathcal{E}|| \, ||h(\mathbf{h}_t)||}$$

**until** processing 20M image pairs or the training budget is exhausted

**Output:** $\mathcal{G}_\theta$

---

## A.4 ADDITIONAL RESULTS FOR GENDR

### A.4.1 ABLATION STUDIES

**Comparison between varied prompt extraction strategies**. We compared several prompt extraction strategies (Null, Fixed, DAPE (Wu et al., 2024c), Qwen2.5-7B (Team, 2024)) in Tab. 3 and found their objective metrics are quite similar. However, in visual comparison, prompts with more semantical descriptions bring more details. Fig. 9 illustrates two examples in which enriching prompts bring improved realism.

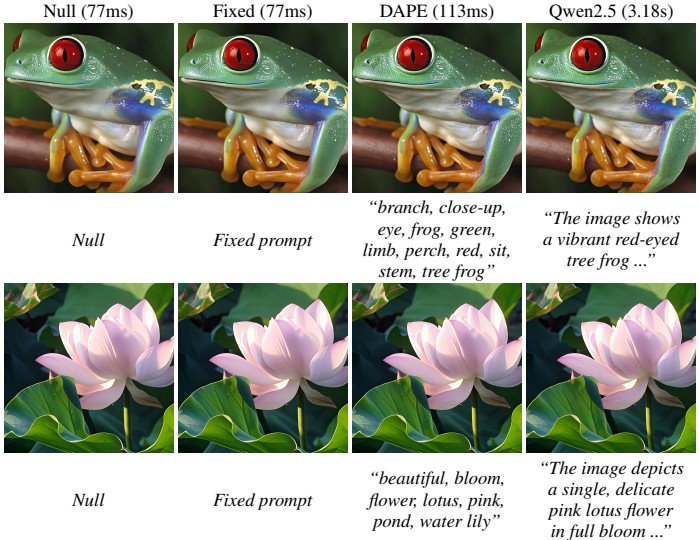

Figure 9: Comparison between images restored by GenDR with different prompt extraction methods.

To evaluate the effectiveness of our strategy for input demanding high semantic control, we test various prompt extraction strategies in the more complex face restoration task. In Fig. 10, we visualize the comparison between fixed prompts with Qwen2.5-VL generated prompt with extremely-degraded face input. Generally, GenDR with a fixed prompt can achieve comparable or even better reconstruction quality than Qwen2.5VL's prompt, demonstrating the effectiveness and generality of the proposed strategy.

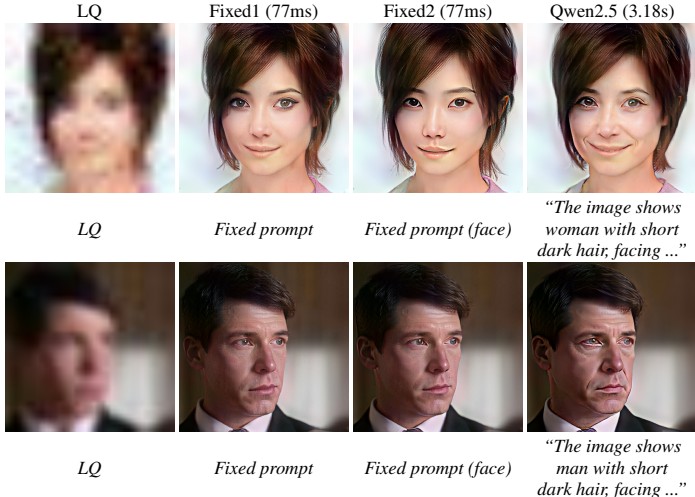

Figure 10: Comparison between images restored by GenDR with different prompt extraction methods in face restoration task. Fixed prompt (face) denotes prompts with face-related descriptions.

**Comparison between varied VAEs**. In Tab. 5 and Fig. 12, we examine the existing VAEs with 4/16 channels. Generally, the 16-channel-based VAEs obtain better reconstruction capability in terms of both QA metrics and subject evaluation. Our VAE is the most lightweight VAE16 that advances existing VAE4 by a large margin and VAE16 from SD3. In Fig. 11, we further visualize their latent distributions via t-SNE. The VAE4 is more uniform than VAEs and reveals clustering. This phenomenon confirms the inferior representation capability of VAE4 than VAE16 since they encode the same 50k images from ImageNet validation. Overall, these results demonstrate our findings that for T2I tasks, VAE16 brings more difficulty in generation but better reconstruction, while VAE4 is more suitable for small models but suffers from detail failure.

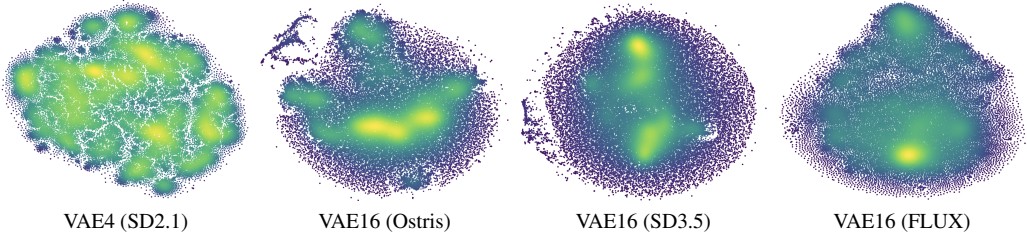

VAE4 (SD2.1)  VAE16 (Ostris)  VAE16 (SD3.5)  VAE16 (FLUX)

Figure 11: Visualization of the latent distribution of varied VAEs.

**Comparison of varied fixed prompt**. We compare several fixed prompts collected from existing work (Yue et al., 2024a; Wu et al., 2024c; Yu et al., 2024a) to find a more effective prompt for local inference embedding. The **positive prompts** are as follows:

- **P1** (InvSR (Yue et al., 2024a)): *"Cinematic, high-contrast, photo-realistic, 8k, ultra HD, meticulous detailing, hyper sharpness, perfect without deformations"*
- **P2** (SUPIR (Yu et al., 2024a)): *"Cinematic, High Contrast, highly detailed, taken using a Canon EOS R camera, hyper detailed, photo-realistic, maximum detail, 32k, Color Grading, ultra HD, extreme meticulous detailing, skin pore detailing, hyper sharpness, perfect without deformations"*
- **P3** (GenDR): *"clean, high-resolution, best quality, smooth plain area, high-fidelity, clear edge, realistic detailed, 8k, perfect without deformations"*
- **P4** (GenDR): *"realism photo, best quality, realistic detailed, clean, high-resolution, best quality, smooth plain area, high-fidelity, clear edge, clean details without messy patterns, high-resolution, no noise, high-fidelity, clear and sharp edge, 4K, 8k, perfect without deformations, great details, 8K, photo taken in the style of Canon EOS –style raw"*

The **negative prompts** are as follows:

- **N1** (InvSR (Yue et al., 2024a)): *"Low quality, blurring, jpeg artifacts, deformed, over-smooth, cartoon, noisy, painting, drawing, sketch, oil painting"*
- **N2** (SUPIR (Yu et al., 2024a)): *"painting, oil painting, illustration, drawing, art, sketch, oil painting, cartoon, CG Style, 3D render, unreal engine, blurring, dirty, messy, worst quality, low quality, frames, watermark, signature, jpeg artifacts, deformed, low-res, over-smooth"*

Table 5: Quantitative comparison between various VAEs. The best and running-up methods are highlighted in **bold** and underlined.

| VAE from | SD2.1 | SDXL | SD3.5 | FLUX | Ours⋆ |
|---|---|---|---|---|---|
| VAE Channel | 4 | 4 | 16 | 16 | 16 |
| #Params | **0.9B** | 2.6B | 8B | 12B | **0.9B** |
| #Params (VAE) | 83.65M | 83.65M | 83.65M | 83.65M | **57.27M** |
| PSNR (dB) | 27.20 | 27.69 | 30.67 | **32.30** | 30.94 |
| SSIM | 0.7857 | 0.8021 | 0.8952 | **0.9214** | 0.8964 |
| LPIPS | 0.1503 | 0.1466 | 0.0833 | **0.0702** | 0.0752 |

⋆: derived from Ostris's vae-kl-f8-d16

Figure 12: Visual comparison between various VAEs. Detail distortions arise in faces and texts for VAE4. (Zoom-in for best view.)

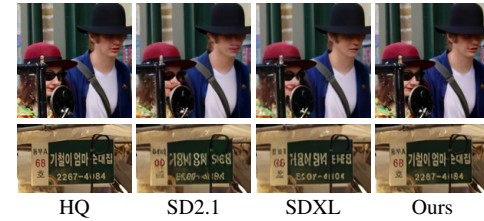

HQ  SD2.1  SDXL  Ours

Table 6: Quantitative comparison (average ClipIQA and MUSIQ) on RealLR between varied fixed (**P**ositive/**N**egative) prompts with CFG=1.2.

|  | Null | P1 | P2 | P3 | P4 | Average |
|---|---|---|---|---|---|---|
| Null | 0.7243/70.62 | 0.7234/71.61 | 0.7118/71.35 | 0.7173/71.77 | 0.7195/71.67 | 0.7193/71.40 |
| N1 | 0.7240/70.58 | 0.7274/71.67 | 0.7145/71.43 | 0.7225/71.83 | 0.7243/71.73 | 0.7225/71.45 |
| N2 | 0.7232/70.45 | 0.7252/71.30 | 0.7128/71.26 | 0.7200/71.70 | 0.7223/71.60 | 0.7207/71.26 |
| N3 | **0.7277**/70.82 | 0.7274/71.77 | 0.7140/71.54 | 0.7224/**71.93** | 0.7242/71.83 | **0.7231/71.58** |
| Average | 0.7248/70.62 | **0.7259**/71.59 | 0.7133/71.40 | 0.7206/**71.81** | 0.7226/71.71 | *0.7214/71.42* |

Table 7: Quantitative comparison (average PSNR, SSIM, LPIPS, ClipIQA, MUSIQ, LIQE) on RealSR with varied CFG factors.

| CFG | PSNR | SSIM | LPIPS | ClipIQA | MUSIQ | LIQE |
|---|---|---|---|---|---|---|
| 1.0 | **22.78** | **0.7009** | **0.2836** | 0.7012 | 68.13 | 4.1690 |
| 1.1 | 22.67 | 0.6978 | 0.2872 | 0.7023 | 68.27 | 4.1909 |
| 1.2 | 22.55 | 0.6946 | 0.2908 | **0.7027** | 68.38 | 4.2076 |
| 1.3 | 22.43 | 0.6912 | 0.2945 | 0.7023 | 68.43 | 4.2172 |
| 1.4 | 22.31 | 0.6877 | 0.2983 | 0.7021 | **68.46** | 4.2225 |
| 1.5 | 22.31 | 0.6801 | 0.2983 | 0.7021 | 68.46 | **4.2227** |
| 1.6 | 22.31 | 0.6801 | 0.3060 | 0.7007 | 68.44 | 4.2147 |
| 1.7 | 21.92 | 0.6761 | 0.3099 | 0.6989 | 68.39 | 4.2004 |

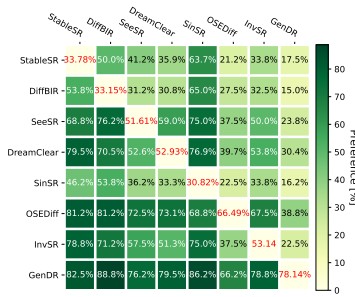

Figure 13: DepictQA preference.

Table 8: Perceptual ablation study on step distillation strategy and loss terms. We omit perceptual and MSE term of 3-7 row for simimplification.

| Loss | RealSR | | | | RealSet80 | |
|---|---|---|---|---|---|---|
|  | PSNR | LPIPS | ClipIQA | MUSIQ | ClipIQA | MUSIQ |
| OSEDiff | 24.57 | 0.3036 | 0.6829 | 67.30 | 0.7037 | 69.19 |
| $\mathcal{L}^{(mse)} + \mathcal{L}^{(per)}$ | 26.72 | 0.2494 | 0.5235 | 58.19 | 0.5806 | 60.50 |
| $\mathcal{L}^{(adv)}$ | 26.40 | 0.2731 | 0.6187 | 62.58 | 0.6544 | 65.10 |
| $\mathcal{L}^{(vsd)}$ | 24.79 | 0.2544 | 0.6496 | 65.01 | 0.6911 | 68.82 |
| $\mathcal{L}^{(sid)}$ | 24.88 | 0.2649 | 0.6511 | 65.39 | 0.7016 | 69.33 |
| $\mathcal{L}^{(cid)}$ | 24.65 | 0.2527 | 0.6532 | 66.85 | 0.7150 | 70.61 |
| $\mathcal{L}^{(cid)} + \mathcal{L}^{(adv)}$ | 22.62 | 0.2899 | 0.7020 | 68.30 | 0.7416 | 71.49 |
| $\mathcal{L}^{(cid)} + \mathcal{L}^{(adv)} + \mathcal{L}^{(repa)}$ | 23.18 | 0.2859 | 0.7014 | 68.36 | 0.7424 | 71.57 |

- **N3** (GenDR): *"noise, blur, oil painting, dotted, compressed, painting, aliased edge, low quality, over-sharp, low-resolution, over-smooth, jpeg artifacts, low quality, normal quality, dirty, messy"*

As shown in Tab. 6, the highest ClipIQA and MUSIQ are obtained with Null-N3 and P3-N3, respectively. We also examine the positive/negative prompts individually. In summary, P1 leads to better ClipIQA while P3 induces advanced MUSIQ. For negative prompts, the proposed N3 outperforms other settings in both attributes of ClipIQA and MUSIQ.

**Comparison of varied CFG**. Referring to InvSR, we implement CFG with a relatively low coefficient, *i.e.*, 1.0 to 2.0. In Tab. 7, we progressively add the CFD factor to 1.7. With increasing CFG, the FR-IQA metrics continue to drop while the NR-IQA first increases and then drops.

**Extensive comparison of varied distillation strategies and loss**. We include more perceptual quality comparisons for the proposed CiDA. As illustrated in Tab. 8, we compare GenDR training under various distillation loss with OSEDiff (Wu et al., 2024b) and ablate each loss term separately. Generally, the proposed CiD achieves better fidelity than VSD (Wang et al., 2024d) and SiD. Compared to OSEDiff, GenDR with CiD maintains similar NF-IQA scores but a higher LPIPS score, illustrating the effectiveness of CiD in consistency preserving. We also conduct ablation study to comprehensively figure out the effectivenss of each loss term. Generally, adding adversarial loss will introduce a perceptual performance (LPIPS) drop but also bring huge no-reference improvement.

### A.4.2 TRAINING AND INFERENCE PERFORMANCE

**Training cost**. To comprehensively evaluate the practical cost for GenDR training with CiD/CiDA, we provide the theoretical/practical training cost (memory/time) in the same setting in Tab. 9.

- **Memory cost**: In vanilla VSD, the fake score network is fully trained, which requires updating two UNets (one-step generator and fake score network) and saving three UNets (with an additional real score network). CiD needs to update three UNets (plus extra discriminator headers for CiDA). We address this by sharing the base model and using switchable LoRA—this preserves only two UNets (one-step generator and shared score network) and optimizes fewer parameters, making CiD/CiDA feasible.

- **Time cost**: In terms of UNet forward passes, VSD/SiD executes 1 pass for generation, 2 passes for score prediction, and 1 pass for fake score updating. CiD adds an extra pass for real score network optimization, while CiDA further adds 1 more pass for GAN loss computation.

- **Practical cost**: We evaluate real training performance as follows. By using ZeRO, LoRA, and sharing strategy, CiD/CiDA requires GPU memory comparable to VSD and SiD, but slightly slows training steps by acceptable 34% and 42%.

Table 9: Training cost comparison of varied distillation strategies. "†" represents using LoRA.

| Method | Baseline | VSD | SiD | CiD | CiD$^\dagger$ | CiDA$^\dagger$ |
|---|---|---|---|---|---|---|
| Forward times | 1 | 4 | 4 | 4 | 5 | 6 |
| Additional trainable parameters (M) | 0 | 865 | 865 | 1730 | 128 | 135 |
| #Time per iteration (s, bs=8) | 1.14 | 3.13 | 3.27 | 4.42 | 4.38 | 4.65 |
| #Memory (GB, bs=8) | 40.12 | 52.81 | 53.38 | 65.57 | 50.51 | 53.44 |

**Inference cost**. We transferred the PyTorch weight of GenDR to TensorRT fp16/int8/fp8 and deployed it on A10 and L20 for practical real-world deployment of server-side enhancement. In the following Tab. 10, we exhibit the actual performance of the overall pipeline at $1024 \times 1024$ and $1440 \times 1440$. Specifically, for 1080p input images ($1920 \times 1080$ has similar pixels as $1440 \times 1440$), GenDR can accomplish real-time restoration with throughputs of $2 \times 1.5$ images/s on A10 and $3 \times 2.6$ images/s on L20 ($3 \times 2.6$ means one L20-48G can deploy 3 GenDR tasks).

Table 10: Inference cost comparison of GenDR deploying on varied GPUs.

| Device-dtype | A100-fp16 | A10-fp16 | A10-int8 | L20-fp16 | L20-int8 | L20-fp8 |
|---|---|---|---|---|---|---|
| Runtime@$1024 \times 1024$, $\approx$720p (ms) | 262 | 305 | 269 | 179 | 157 | 160 |
| Runtime@$1440 \times 1440$, $\approx$1080p (ms) | 601 | 721 | 627 | 424 | 369 | 381 |
| Memory@$1920 \times 1440$ (GB) | 13.27 | 12.53 | 12.53 | 12.60 | 12.60 | 12.60 |

### A.4.3 MLLM ARENA

In Fig. 13, we utilize DepictQA to make pairwise comparisons and calculate the average selected rate (in red). The GenDR receives about 78% votes, indicating its better quality than other methods.

### A.4.4 VISUAL RESULTS

In Fig. 16, we provide more visual comparisons on RealSet80 (Yue et al., 2024b) between the proposed GenDR and other diffusion models. Generally, the GenDR can reasonably preserve the original information and generate more faithful details. In Fig. 17, we compare our method with OSEDiff and RealESRGAN on the KwaiSR dataset for UGC (user-generated content) SR Li et al. (2025). The proposed GenDR provides faithful reconstruction and the highest fine-grained detail restoration.

### A.5 PRIMARY RESULTS FOR GENDR DISTILLED FROM SD3.5

To demonstrate the generality and expansion of GenDR, we train an SDXL-based GenDR and employ SD3.5-large as a distillation teacher. Specifically, we change the 4-channel latent space and

diffusion scheduler of a pruned SDXL (1.3B) to a 16-channel VAE space of SD3.5 and flow matching scheduler, and then use SD3.5 as fake/real prediction network of CiD to conduct SR distillation for SDXL-VAE16. The process and results of this stage are summarized in Fig. 14: SDXL-VAE16 adopts SD3's 16-channel VAE, reduces parameters by half compared to the original SDXL, while achieving restoration quality comparable to the baseline.

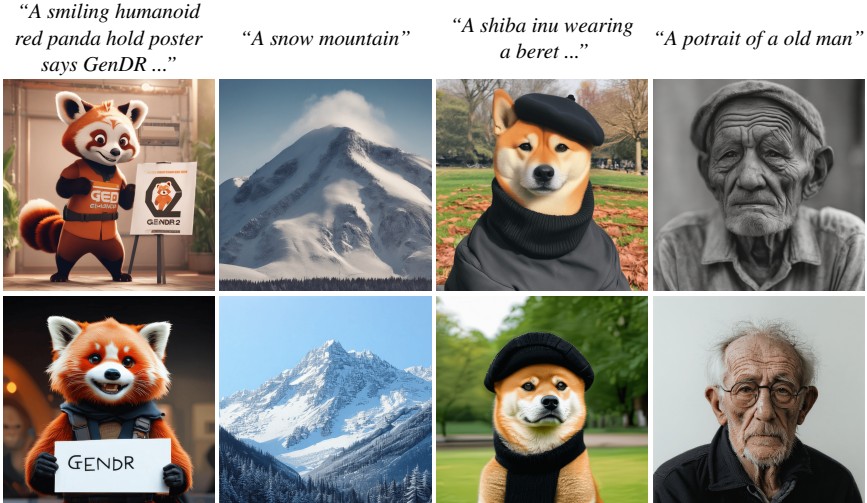

Figure 14: Samples produced by SDXL (Rombach et al., 2022) (**Above**, 2.6B Diffusion Model with 4-channel VAE) and SDXL-VAE16 (**Bottom**, VAE16, 1.3B Flow Model with 16-channel VAE).

Through using SDXL-VAE16 as generator $\mathcal{G}_\theta$ and SD3.5-large as initial $f_\psi$ and $f_\phi$, we use a flow-matching adapted Eq. (9) to train primary SDXL-based GenDR, termed GenDR-1.3B. We provide the primary results of the 10k iteration training in Fig. 15.

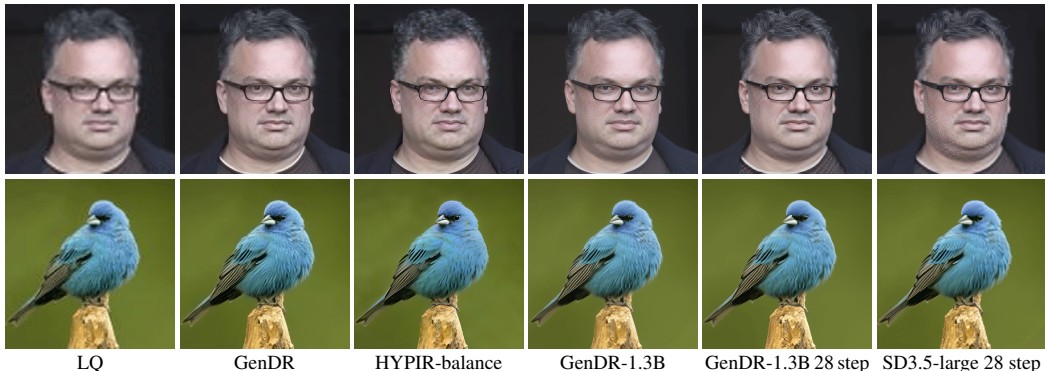

| LQ | GenDR | HYPIR-balance | GenDR-1.3B | GenDR-1.3B 28 step | SD3.5-large 28 step |

Figure 15: Samples produced by GenDR, HYPIR (Lin et al., 2025), SDXL-based GenDR, and SD3.5-large with ControlNeXt (Peng et al., 2024).

## A.6  ADDITIONAL RESULTS FOR SD2.1-VAE16

### A.6.1  QUANTITATIVE REUSLTS

To further verify the usability of SD2.1-VAE16, we provide detailed results of GenEval (Ghosh et al., 2023) and additional CLIP results in Tab. 4. Generally, the proposed SD2.1-VAE16 is slightly lagged behind SD2.1 (similar CLIP but lower GenEval), which is aligned with our expectation since increasing the latent space dimension will introduce a generation difficulty (Esser et al., 2024).

Table 11: Comparison of T2I methods.

| Model | #Params | GenEval↑ | CLIP↑ |
|---|---|---|---|
| PixArt-$\alpha$ | 0.6B | 0.48 | 0.316 |
| SD1.5 | 0.9B | 0.43 | 0.322 |
| SD2.1 | 0.9B | 0.50 | 0.30 |
| SD-Turbo | 0.9B | - | 0.335 |
| SDXL | 2.6B | **0.55** | **0.343** |
| SD2.1-VAE16 | 0.9B | 0.48 | 0.323 |

### A.6.2 Qualitative Results

In Fig. 18, we show the images generated by the proposed SD2.1-VAE16 with the following prompts:

- An astronaut riding a horse in the forest.
- A girl examining an ammonite fossil.
- A chimpanzee sitting on a wooden bench.
- A penguin standing on a sidewalk.
- A goat wearing headphones.
- Dreamy puppy surrounded by floating bubbles.
- An elephant walking on the Great Wall.
- An oil painting of two rabbits in the style of American Gothic, wearing the same clothes as in the original.
- A woman wearing a red scarf.
- Motion.
- A squirrel driving a toy car.
- A giant gorilla at the top of the Empire State Building.
- A bowl with rice, broccoli and a purple relish.
- Stars, water, brilliantly, gorgeous large scale scene, a little girl, in the style of dreamy realism.
- A cat reading a newspaper.
- A woman wearing a cowboy hat face to face with a horse
- A watermelon chair.
- A photo of llama wearing sunglasses standing on the deck of a spaceship with the Earth in the background.
- Pirate ship trapped in a cosmic maelstrom nebula, rendered in cosmic beach whirlpool engine, volumetric lighting, spectacular, ambient lights, light pollution, cinematic atmosphere, art nouveau style, illustration art artwork by SenseiJaye, intricate detail.

## B    The Use of Large Language Models (LLMs)

We use Large Language Models in this research project to assist with various aspects of the writing and research process, including text polishing and refinement, and grammar enhancement.

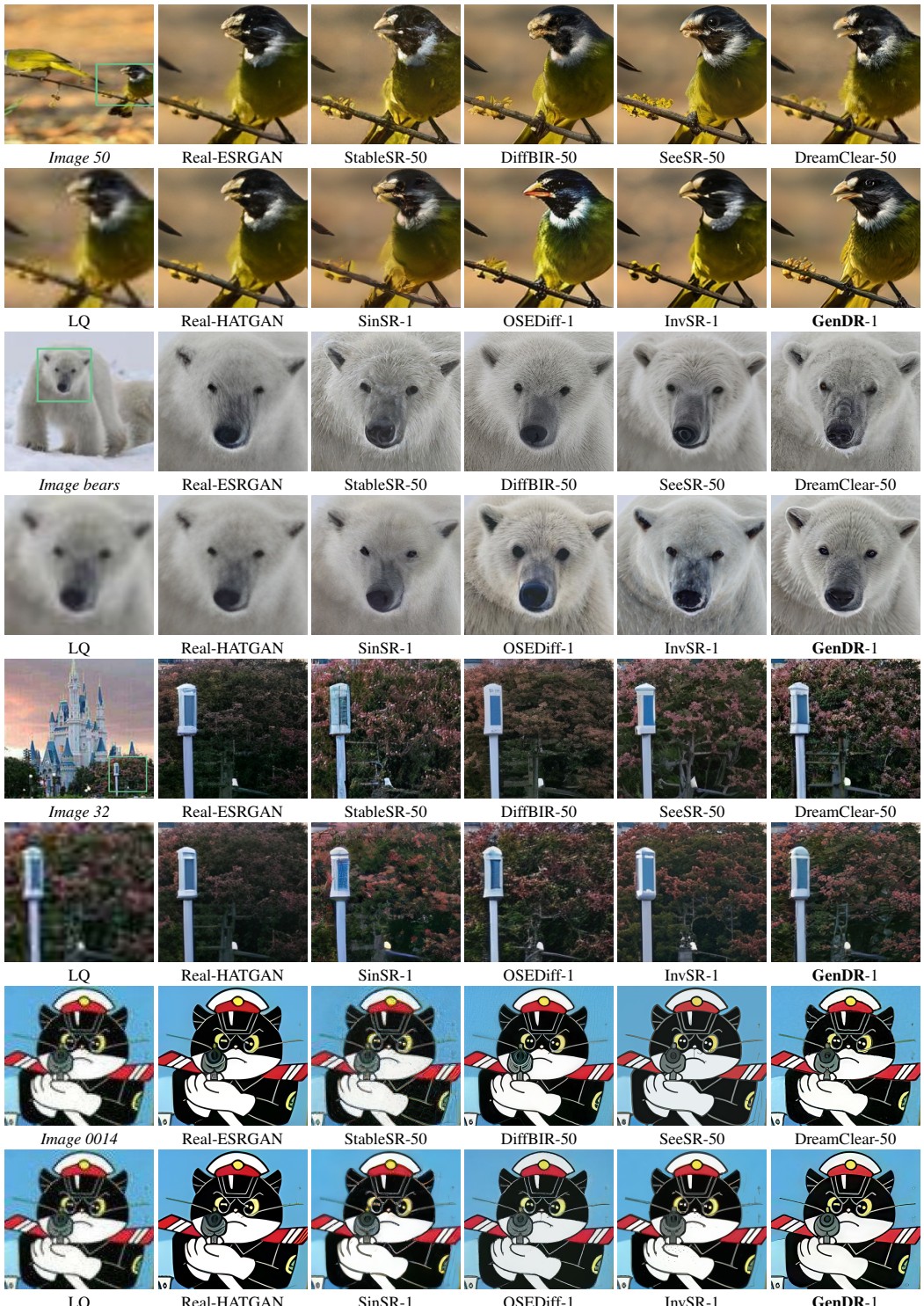

Figure 16: Visual comparison of GenDR with other methods for ×4 task. (Zoom-in for best view.)

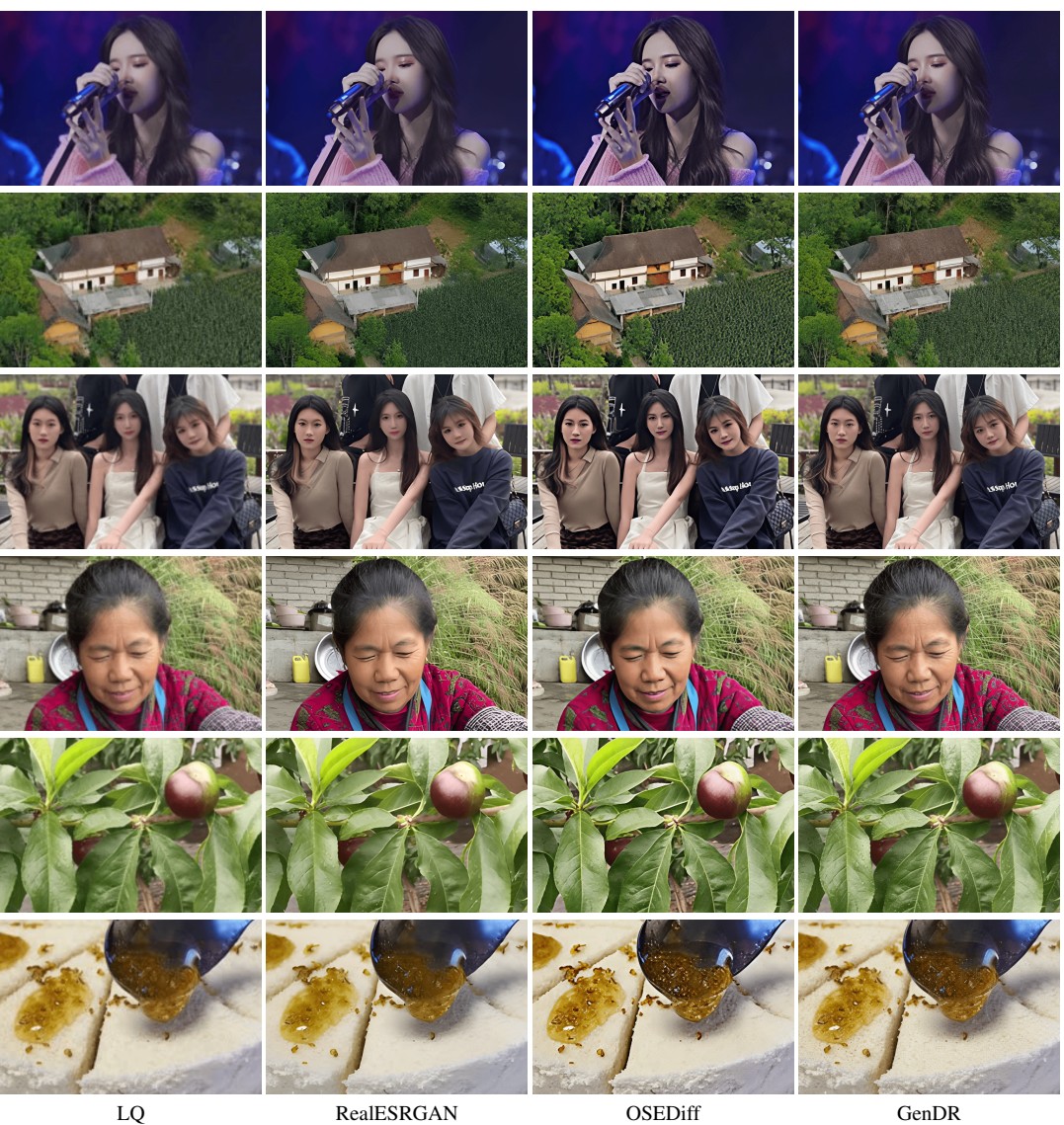

| LQ | RealESRGAN | OSEDiff | GenDR |

Figure 17: Visual comparison of GenDR with other methods for UGCSR. (Zoom in for best view.)

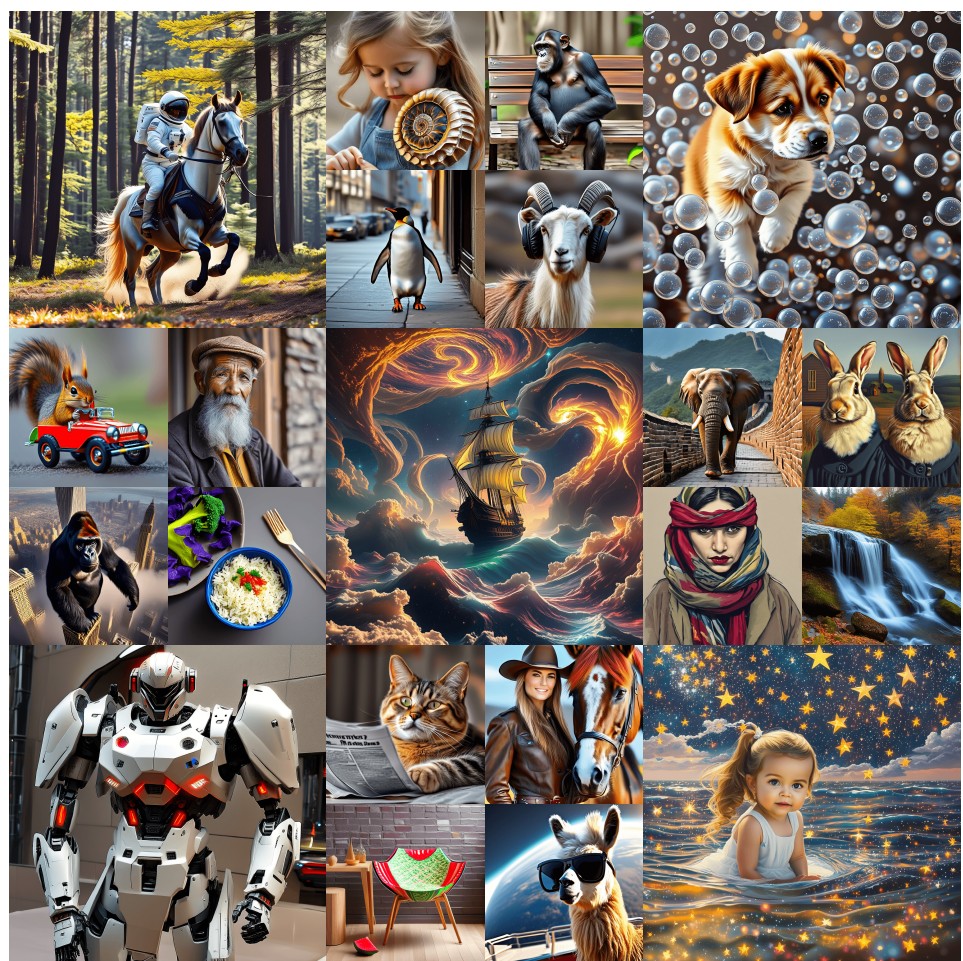

Figure 18: $1024^2$px and $512^2$px samples produced by SD2.1-VAE16.

