# OpenReview forum: "GenDR: Lighten Generative Detail Restoration"
_ICLR.cc/2026/Conference — ICLR 2026 Poster_

### Official Review · Reviewer_kCw1 · 2025-10-21

**Soundness:** 3
**Presentation:** 3
**Contribution:** 2
**Rating:** 6
**Confidence:** 4

**Summary:**

This paper proposes GenDR for better tradeoff between detail enhancement and inference efficiency. By utilizing a pre-trained 16-channel VAE, it expands the restoration capacity of the proposed SR model with a 0.9B UNet backbone. To reduce computational overhead, it introduces consistent score identity distillation technique to effectively train a one-step model while preserving its ability to generate vivid details. Extensive experiments demonstrate its ability on diverse scenarios.

**Strengths:**

* By utilizing a pre-trained 16-channel VAE, GenDR effectively expands the capacity of the proposed SR model.
* It proposes a SR-tailored step distillation technique CiDA that restore vivid details and stabilize training.
* Extensive experiments demonstrate its effectiveness on various benchmark.

**Weaknesses:**

* Since one of the main contributions of this paper is expanding SR capacity by a large channel VAE, more analysis about this component should be added, including the choice of the latent channel, training difficulty and reconstruction ability.
* Results of multi-step GenDR model are absent. Although authors provide comparison between various distillation methods, comparing with the teacher model can further validate the performance of the proposed CiDA.
* While replacing the text encoder by a fixed positive embedding reduces inference overhead with slight performance degradation, results on Tab. 7 shows that slightly increasing CFG scale leads to worse performance. I am concerned that the weak textual guidance might come from using a fixed embedding. The authors should consider testing on input images that require high semantic control, such as portraits or text-containing inputs.

**Questions:**

* The proposed method utilizes a generator and two trainable score networks to perform the score distillation process. Thus, quantitative comparison of memory usage should be provided.
* The authors should consider providing more image results as supplemental material.

---

> ### Author Response · Authors · 2025-11-18
>
> Thank you for constructive feedback.  We try to address the concerns in both the weaknesses and the questions, respectively.
>
> 1. **About more discussion for VAE**.  We provide more comparisons for varied VAE in Fig. 11 and Tab. 5. In general, we select the VAE channel to be 16 for two reasons.
> - T2I prior. Existing T2I tasks have shown that the 16-channel VAE achieves better trade-offs between reconstruction and generation quality. As shown in the following table, we compare VAE with channels 4, 8, and 16, where the 16-channel VAE has the best reconstruction quality. We didn't extend our search range to channel 32 or 64 since our CiD is still founded on T2I training, limited by the T2I bound. The recent DiT work can achieve about 1.0 gFID, which is close to rFID. As we know larger VAE latent space represents harder generation training, that is, lower gFID,  it is ineffective to further improve the reconstruction quality of VAE when gFID > rFID.
>
>     | VAE channel | 4 | 8 | 16 |
>     | :- | :-: | :-: | :-: |
>     | rFID |  2.41 | 1.56 | 1.06 |
>     | PSNR |  25.12 |  26.40 | 28.62 |
>
> - Generality. Most existing well-pretrained T2I DiT is based on a 16-channel VAE; we can extend our results for cross-architecture distillation. For example, we distill SDXL with SD3.5 in section A.5.
>
> 2. **About comparison with multi-step GenDR**. We didn't include the comparison with multi-step GenDR since the GenDR is directly distilled from T2I model (SD2.1-VAE16), similar to OSEDiff distilled from SD2.1. However,  we enable training of a real score network, whose training target is a multi-step SR model to ensure consistency. Based on the real score network, we implement a multiple-step SR diffusion model. We provide our latest results on the SD3.5 and SDXL models in the following table. Generally, 1-step GenDR is slightly behind 28-step GenDR, but the gap is acceptable. In Fig. 15, we provide visual comparison, where their visual quality is comparable. More details can be found in section A.5 of the revised manuscript.
>
>    | Method | GenDR (paper) | GenDR | GenDR 28-step |
>    | :-: | :-: | :-: | :-: |
>    | ClipIQA | 0.6543 | 0.6675 | 0.6792 |
>    | MUSIQ | 66.16 | 68.31 | 70.31 |
>
> 3. **About removing the text encoder in more complex scenes**. We try to address the concern through both analysis and experiment. Generally, the CFG has a huge effect on multiple-step inference. However, in one-step inference, increasing the CFG will directly and sensitively influence the output latent, which would amplify the generation error and bring artifacts, especially for high-frequency details like edges. This process is similar to RGB overflow. We also evaluate the effectiveness of removing the textual branch in the face restoration task. In Fig. 10, we compare the results produced with the fixed prompt and the Qwen2.5-VL-generated prompt under CFG=1.2. Specifically, GenDR with a fixed prompt can achieve comparable performance. Moreover, in the second case of Fig. 10, the Qwen2.5 prompt encounters the "overflow" phenomenon as mentioned above, which aligns with our results that increasing CFG scale (even a small CFG factor) might result in artifacts of final results for one-step model.
>
>
> 4. **About training and inference memory cost**. In Tables 9 and 10 of our revised manuscript, we provide a detailed quantitative comparison and analysis of memory usage and latency for both training and inference. In vanilla VSD, the fake score network is fully trained, which requires updating two UNets (one-step generator and fake score network) and saving three UNets (with an additional real score network). CiD needs to update three UNets (plus extra discriminator headers for CiDA). We address this by sharing the base model and using switchable LoRA—this preserves only two UNets(one-step generator and shared score network) and optimizes fewer parameters, making CiD/CiDA feasible. In detail, GenDR with CiD/CiDA requires only 50.51/53.44GB for batchsize=8 training, which is comparable with VSD (52.81GB).
>
> 5. **About more visual comparison**. We added more visual comparisons for UGC content in our revised submission, and will add more visual comparisons as suggested. We are arranging the manuscript to make the size under OpenReview restriction.

---

### Official Review · Reviewer_5Lfg · 2025-10-30

**Soundness:** 2
**Presentation:** 1
**Contribution:** 2
**Rating:** 4
**Confidence:** 4

**Summary:**

This paper presents GenDR, a method for efficient super-resolution (SR) that combines VAE-16 and small latent diffusion models (LDMs) using score distillation. The authors claim their approach enhances restoration quality while maintaining speed, achieving superior performance over existing methods. The paper’s main focus appears to be on improving the balance between fidelity and inference time in diffusion-based SR models.

**Strengths:**

1. The combination of VAE-16 with latent diffusion models, coupled with score distillation, is an interesting approach for super-resolution.
2. The experimental results demonstrate that GenDR outperforms existing models in terms of both quality and efficiency, with strong quantitative and qualitative performance across multiple benchmarks.
3. The method is designed to be efficient, offering improved restoration speed without compromising visual fidelity, which is a significant advantage in real-world applications.

**Weaknesses:**

1. The paper struggles with clarity and coherence, especially in how it connects its contributions. While the abstract and introduction spend significant time discussing the advantages of using VAE-16 for SR tasks, the method section shifts focus entirely to the proposed CiDA loss. This abrupt shift makes it difficult to understand the relationship between the two contributions, as they don’t seem to be closely tied.
2. While the paper introduces CiDA as a novel technique for distilling scores, its relevance to the task of detail restoration in SR is not sufficiently discussed. The paper does not clearly explain why this loss function is needed for SR or how it helps improve the restoration of fine details. The overall discussion of CiDA feels more aligned with general diffusion model training rather than specifically addressing the SR challenge.
3. The methodology section is dense and difficult to follow. The authors jump between various components like VAE-16, CiD, and adversarial learning without fully explaining their relationships or how they work together to address the SR problem. Additionally, the notations used in the equations make the explanation harder to follow (e.g, $f(\cdot)$ and $\epsilon(\cdot)$ seem to both refer to the scores).

**Questions:**

See above.

---

> ### Author Response · Authors · 2025-11-18
>
> We appreciate the reviewer's careful and explicit feedback, and we would like to reclarify our motivations and methods to help comprehensively understand the coherence between VAE and distillation and relevance for CiDA to the SR task.
>
> 1. **About coherence for 16-channel VAE and 1-step inference**. In our original abstract and introduction, we discuss both VAE and distillation, and we try to reoutline the process. Begin with our motivation, as highlighted in *L064-067*, "two problems persist that restrict the practical usage of diffusion-based SR: slow inference speed and inferior detail fidelity.", we try to solve these problems systematically by revisiting the difference between the diffusion-based T2I task and SR task. In *L085-095*, we show that, 1) compared T2I task, the SR task focuses more on detail, thus needing higher higher-dimensional latent space, i.e., 16-channel VAE; 2) the SR task only restores the lost high-frequency details, generating less information than the T2I task, thus enabling a shorter diffusion path or efficient model, i.e., 1-step diffusion on UNet. Based on these two insights, we summarize and discuss our contribution to both VAE and distillation in *L096-114*. Generally, the usage of 16-channel VAE and 1-step distillation (CiD) is closely tied, especially since the existing T2I task has validated that a higher-dimensional latent space requires a larger model, and GenDR provides a new insight into the SR task that the SR task needs one-step diffusion under high-dimensional latent space.
>
> 2. **About the relevance of CiD/CiDA to the SR task**. We try to rearrange the relevance of both CiD and CiDA to the SR task.
>
> - CiD: In *L202-209*, we point out that the existing distillation methods (VSD/SiD) are tailored for the T2I task and would raise trouble for the SR task since they share distinct targets and varying training distributions, leading to quality and content inconsistency. To prevent the inconsistency, existing methods utilize the regression loss between the GT and SR images. In contrast, we develop an SR-tailored distillation strategy by adjusting the real score network and introducing regression loss of SR into SiD to eradicate these inconsistency problems.
>
> - CiDA: The key motivation to develop CiD to CiDA is to improve generation realism and accelerate training. We acknowledge that despite retraining for image restoration tasks, SD2.1-VAE16 may still suffer from "T2I-generated fakeness" due to:
>
>    - Availability of high-quality image data for training is limited, which restricts the model's ability to fully learn the authentic characteristics of real-world images.
>    - SD2.1-VAE16 is inherited from the T2I model, and some of the T2I model's inherent tendencies, such as generating content with a certain degree of artificiality to meet text prompts, may persist.
>
>    Despite CiD providing better consistency than VSD and SiD, it still relies on a diffusion prior and mainly focuses on aligning the training target and leveraging SR priors through score distillation, but it does not specifically address the generated fakeness. To tackle this, we introduce adversarial learning into CiD to adversarially push the generated latent towards the manifold of real high-quality images. On the other hand, since SD2.1-VAE and previous work have proven that representation alignment can help convergence, we continue using the alignment of SD2.1-VAE16 to accelerate training.
>
> 3. **About the methodology clarity**. In the methodology section, we follow a logical, interdependent chain tailored to SR’s core needs (detail preservation + efficiency).  We rehash the method section according to the manuscript for better comprehension:
>
>    - VAE-16 provides a high-dimensional latent space to retain fine HR details (solving 4-channel VAE’s detail loss). It supplies LR/HR latents for all subsequent steps.
>    - CiD develops SR-specific score distillation that uses VAE-16’s HR latents as identity targets, ensuring consistency between LR and generated SR.
>    - CiDA enhances CiD with adversarial learning and representation alignment (on VAE-16’s latents) to boost realism and training stability.
>    - Simplified Pipeline integrates VAE-16 and CiDA-distilled UNet into a one-step framework for fast SR inference.
>
>
>    Since the main theoretical contribution is CiD/CiDA, we provide a detailed mathematical analysis of how we develop VSD into CiDA, which might make the method section dense. We believe that all symbols are explicitly defined. As to mentioned $f(\cdot)$ and $\epsilon(\cdot)$, they do denote the noise-predicted based  (defined in L192)  and latent-predicted score (defined in L238) functions and can be equivalently transformed by $ f(t) = \frac{ f(0) - \bar{\beta}_t \epsilon(t)}{ \bar{\alpha}_t}$.

---

> ### Comment · Reviewer_5Lfg · 2025-11-25
>
> Thank you for the response.
>
> However, the confirmation that $f(\cdot)$ and $\epsilon(\cdot)$ are interchangeable only reinforces my concern that the methodology is unnecessarily convoluted.
>
> Firstly, the mathematical derivation of CiD is confusing and mathematically redundant. Switching notation from $\epsilon$-space in Eq. (5) to $f$-space in Eq. (8) forces the reader to track a trivial parameterization change that adds no theoretical value. This is particularly perplexing given that your experiments rely on SD2.1, which is natively a noise-prediction model. Converting the formulation to $f$-space is counter-intuitive and seems designed to inflate the complexity of the method rather than clarify it.
>
> Secondly, the dense mathematical description obscures the fact that the **only** substantive contribution of the CiD loss is the substitution of the generated latent $z_g$ with the ground-truth latent $z_h$. This is essentially a standard supervised guidance signal anchored to the SR ground truth. By wrapping this simple substitution in a change of prediction space, the paper frames a straightforward concept as a complex derivation, which is redundant and obscuring.
>
> Additionally, while I understand the motivation for the 16-channel VAE, the paper still lacks a rigorous theoretical analysis justifying this specific architectural choice beyond empirical observation, which is a significant omission given it is a core contribution intended to solve the fidelity dilemma, as also raised by reviewer kCw1.
>
> Finally, I am also concerned that GenDR demonstrates a notable drop in established fidelity metrics like PSNR and LPIPS compared to baselines like Real-HATGAN and DiffBIR in Table 1, which contradicts the claim of achieving superior detail fidelity, which is also mentioned by reviewer Pt8i.
>
> Overall, the writing of this paper lacks the polish and needs substantial revision. I am lowering my rating and demand that these issues be resolved before any other consideration.

---

> ### Author Response · Authors · 2025-11-25
>
> Thanks for your feedback. We regret that our rebuttal cannot address your concern. We hope to reclarify the mentioned question.
>
> We notice the reviewer's main concern with the method part is due to the misunderstanding of $\epsilon(\cdot)$ and $f(\cdot)$. We hope to clarify why we need these two forms of score in our paper. In the previous multi-step T2I-based model, most of them, like SD2.1,  predict noise $\epsilon$; thus, the distillation function is based on noise. However, for a one-step SR model, the prediction of noise is redundant since only end-to-end (one step) inference is executed, and the noise is directly transferred to the latent according to Eq (2); that is, one-step SR is more intuitively regarded as an $f$-space map between LR and SR. Another reason is that using $\epsilon$ is unable to conduct an identity transformation for $z_g$ because there doesn't exist a proper estimated $\epsilon$ for $z_h$. Thus, converting the formulation to latent space is essential.
>
> We try to dispel your other concerns:
>
> 1. About the mathematical derivation of CiD is confusing and mathematically redundant. Actually, all derivations are necessary. It is essential to show the limitations of using existing T2I-based distillation in the SR task and thereby conveying why we need to realign the distillation function to the SR target. Moreover, we think directly exhibiting Eq. (7)
>  and (8) would bring more confusion for the reader without distillation priors. The reviewer emphasizes the mathematical redundancy but provides no example, which confuses us a lot.
>
> 2. About substantive contribution of the CiD. We hope the reviewer can **recheck** the modification of the proposed CiD (Eq (7) and (8)) over the existing VSD and SiD. CiD **not only** substitutes the generated latent $z_g$ with the ground-truth latent $z_h$ **but also** makes both real and fake score functions $f_\phi$ and $f_\psi$ trainable to align with the SR target, thereby ensuring consistency. Hence, it is inappropriate to summarize the CiD as a "simple substitution". Regarding "frames a straightforward concept as a complex derivation", it is not merely a straightforward substitute but supported by mathematical analysis, as we provide the proof of CiD in the supplementary material. We regard the "straightforward concept" as an appraisal that the CiD is intuitive and effective.
>
> 3. About VAE ablation. We update more results as shown in our response to reviewer kCw1. The main reason for not conducting more experiments on VAE is the resources. It is hard to train more VAEs with various latent spaces and then train UNets with them.
>
> 4. About fidelity metrics.  We have addressed the question in our response to reviewer Pt8i. In fact, the mentioned RealHATGAN is the pixel2pixel-based method, which undoubtedly outperforms the diffusion-based method in fidelity. As mentioned in our response to reviewer Pt8i, we try to align the overall fidelity performance at a similar level as OSEDiff and improve subjective performance. Hence, compared to DiffBIR (6213ms), our GenDR (77ms) has a superior BF-IQA score but it also advances 2/3 metrics of FR-IQA, which are listed as follows.
>
>     | Method | PSNR | SSIM | LPIPS |
>     | - | - | - | - |
>     | DiffBIR | 25.45 | 0.6651 | 0.2876 |
>     | GenDR | 24.14 | 0.6878 | 0.2652 |
>
> We are glad to continue discussing these questions.

---

> > ### Comment · Reviewer_5Lfg · 2025-11-28
> >
> > Thank you for the follow-up. I appreciate the clarifications. To better understand the contribution and ensure the methodology is accurately represented, could you please address the following questions regarding the formulation and implementation?
> > 1. Regarding the proof in Appendix A.1, Equation 14 identifies the optimal denoiser $f_{\theta^*}$ as the expected ground truth $\mathbb{E}[z_h|z_t]$, and Equation 15 utilizes this to substitute the generated target $z_g$ with $z_h$.
> >     * Does this derivation primarily serve to formalize the substitution of the distillation target with the ground truth?
> >     * If so, would it be more accurate to describe the method as supervised fine-tuning (anchored to $z_h$) combined with a distillation regularizer, rather than framing it as a dense derivation of a new distillation paradigm?
> > 2. I notice that Algorithm 2 treats the pretrained backbone as $f_\mu$ and computes all loss terms ( $L_\phi, {L}_\psi, {L}^{(cida)}$ ) in $f$-space.
> >     * Given that the backbone is Stable Diffusion 2.1 (an $\epsilon$-prediction model), does the implementation implicitly perform a transformation ($\epsilon \to f$) during the forward pass (which is not discussed in the main text nor the appendix)?
> >     * If such a transformation is used successfully in Algorithm 2, does this not suggest that the $\epsilon$ and $f$ spaces are practically interchangeable, contrary to the claim that $\epsilon$-space is unsuitable?
> > 3. There appears to be a confusing mismatch between the terms defined in the text and the implementation shown in the algorithm. Equation 9 in the main text defines the objective as a hybrid combination of ${L}^{(3)}$ (in $f$-space) and ${L}^{(1)}$ (in $\epsilon$-space). However, Algorithm 2 implements the corresponding regularization term $L_\psi$ entirely in $f$-space (  $||f_\psi - \text{sg}[z_g]||^2$  ).
> >     * Could you clarify this discrepancy? The inconsistency between the theoretical formulation and the algorithm makes it unclear which objective was actually optimized in the experiments.
> >     * Would unifying the notation in the text to match the algorithm (all $f$-space) not resolve this confusion?
> >
> >
> > I believe addressing these points would significantly improve the clarity of the paper and better highlight the actual practical contribution.

---

> ### Author Response · Authors · 2025-11-28
>
> Thanks for your feedback, which enables us to understand your concerns clearly. We would help the reviewer to comprehend the mentioned questions.
>
> 1. Yes,  Eq (14) identifies the optimal denoiser $f_{\theta^*}$ as the expected ground truth $\mathbb{E}[z_h|z_t]$ and thereby uses more consistent $z_h$ to replace unstable $z_g$. However, we think it is more proper to regard the entire CiD framework as an SR-tailored distillation paradigm. We summarize two reasons to support our explanation.
>
> - Despite substituting $z_g$ with $z_h$, as shown in Eq. (8), CiD doesn't conduct **strong** supervision (e.g., MSE loss) to directly backward gradient according to $z_h$. Actually, the $z_h$ is *de facto* an **assistant term** to provide **oblique guidance and stabilize training**. To comprehensively understand it in the CiD paradigm, we start with the foundation of score distillation, which optimizes the teacher diffusion model, the student diffusion model, and the student generation/SR model. For simplicity, we unify the loss in $f$-space.
>
>    - (1) Teacher diffusion: $\psi^{\*} = \text{argmin}\mathbb{E}|| f_\phi(z_t) - z_0||^2 $
>    - (2) Student diffusion: $\psi^{\*} = \text{argmin}\mathbb{E}|| f_\psi(z_t^{g}) - z^g_0||^2 $
>    - (3) Student generation/SR:  $\theta^{\*} = \text{argmin}\mathbb{E}|| f_{\phi*}(z_t^{g}) - f_{\psi*}(z_t^{g})||^2 $, where $z_g = f_\theta(z_T)$
>
>    As (1) has been carefully solved in pre-training, to train a one-step generator/SR with (3), we first need to find the optimal solution $\psi^{\*}$ of formula (2) and then update $\theta$ accordingly. However, in practical implementation, **(a)** we have to cyclically update $\psi$ and $\theta$; and **(b)** the $\psi^{\*}$ is regarded as a scalar rather than $\psi^{\*}(\theta)$ for convenience, leading to unstable training as a GAN-based method.
>
>   Although a longer training can ensure $\psi$ closer to $\psi^{\*}$, the dilemma of **(b)** also restricts the stable and consistent training procedure. To alleviate the problem, **SiD/CiD reduces the dependence of $\psi^{\*}$** by identity transformation, that is, using optimal $z_g$ or SR-tailored $z_h$ to replace $\psi^{\*}$. It is a core adaptation of distillation theory rather than simply using ground truth as calibration. Moreover, compared to the original SiD, we resolve the inconsistency of $ \psi^{\*}$ and further integrate the SR prior to utilize $z_h$ replacing $z_g$.
>
>
> - As emphasized in our rebuttal/paper, CiD also includes distillation paradigm modification tailored for SR. As mentioned above, the foundation of the existing distillation strategy is based on the pretraining on the T2I dataset via Formula (1). However, as many researchers have noticed, the T2I dataset focuses more on image-text alignment rather than providing high-quality images, which, in fact, harms Formula (3). Hence, we propose Eq. (7) to solve this inconsistency.
>
>    These construct the core contribution of CiD to the existing T2I-tailored distillation paradigm.
>
> 2. We hope to discuss the detailed implementation of SD2.1 to dispel your concern. For SD2.1 (diffusers), it implicitly performs a transformation from $\epsilon$ to $f$. In fact, for SD2.1/SDXL/SD3,  no matter $\epsilon$-prediction or $v$-prediction, they input $f_t$ and output $\epsilon_t$/$v_t$, and then use scheduler to produce $f_{t-1}$. Thus, it is intuitive to use the $f$-space. As to the second question, although $f$ and $\epsilon$ are practically interchangeable and we can estimate a $\epsilon_h$ for $z_h$, the $\epsilon_h$  is a handmade value without a comprehensive physical definition, which is unsuitable compared to the original $z_h$, which can be regarded as ground truth to help understanding.
>
> 3. Yes, they should be in $f$-space. We will correct it in our revised paper. As for unifying all formulations in $f$-space, it is a good idea, but the denoise process is also representative, and some related methods are $\epsilon$-based. We will take it into consideration and plan to unify them in our method part.
>
> Thanks again for your involvement in the rebuttal period. We are glad to continue discussing these questions.

---

### Official Review · Reviewer_Pt8i · 2025-10-31

**Soundness:** 3
**Presentation:** 2
**Contribution:** 3
**Rating:** 6
**Confidence:** 4

**Summary:**

This paper proposes GenDR, a diffusion-based model for real-world image super-resolution (SR) that aims to address the long-standing trade-off between detail fidelity and inference efficiency. The authors identify that most text-to-image (T2I) diffusion models are suboptimal for SR tasks because (1) they use low-dimensional latent spaces (typically 4-channel VAEs), and (2) they require multi-step inference to synthesize images. To overcome this, the authors develop an SD2.1 model with a 16-channel VAE backbone and a consistent score identity distillation strategy.  Empirically, GenDR achieves competitive or superior performance over state-of-the-art (SOTA) models such as DiffBIR, OSEDiff, and DreamClear, with notable improvements in LIQE, MUSIQ, and Q-Align scores and a 77 ms runtime.

**Strengths:**

I agree with the claim on the disadvantages of the 16-channel VAE, which probably limits the performance upper bound of diffusion-based SR methods. As illustrated in Table 2, such a modification indeed improves the SR results.

**Weaknesses:**

1. My main concern mainly focuses on the experimental performance. The authors claim that the proposed method is able to enhance the detail fidelity. The quantitative comparison results in Table 1 cannot support such a claim, in which GenDR does not show obvious improvements regarding the fidelity metric, such as PSNR, LPIPS.

2. As for the visual results, I personally don't think GenDR is better than other SoTA methods, particularly in the first example in Fig. 1 and the second example in Fig. 5.

3. As for the efficiency, GenDR shows slight advantages compared with InvSR and OSEDiff. I guess such improvement is due to the removal of the text encoder. However,  this trick is also suitable for other methods (e.g., InvSR) that do not rely on dynamic textual information.

**Questions:**

I wonder about the effectiveness of RAPE regularization in SR task.

---

> ### Author Response · Authors · 2025-11-18
>
> We appreciate the reviewer's informative and constructive feedback. We try to address the concerns in both the weaknesses and the questions, respectively.
>
> 1. **About perceptual quality**.  We agree and understand the reviewer's concern that the fidelity metrics of GenDR (PSNR/LPIPS) in Table 1 are comparable to existing methods. Still, we would like to explain that it is more due to the trade-offs between reconstruction fidelity and detail richness than the invalidation of the proposed CiD.
>
> -  Training preference. GenDR is a subjective-quality-oriented model, which is expected to generate more details to improve visual quality. Thus, during the training stage, we use more loss terms like adversarial loss and a large factor for CiD, and try to align the overall fidelity performance at a similar level as OSEDiff. However, when training under the same settings, the proposed CiD can significantly improve the perceptual quality. As shown in the following table, compared with OSEDiff with VSD/VAE4, the GenDR with VSD/VAE16 achieves 0.03 improvement on LPIPS, showing the effectiveness of 16-channel VAE in fidelity preservation. Moreover, the proposed CiD strategy brings both LPIPS and NR-IQA improvement.
>
>     | Loss | LPIPS | ClipIQA | MUSIQ |
>     | - | - | - | - |
>     | OSEDiff | 0.3036 | 0.6829 | 67.30 |
>     | $\mathcal{L}^{(\text{vsd})}$ | 0.2744 | 0.6496 | 65.01 |
>     | $\mathcal{L}^{(\text{sid})}$ | 0.2649 | 0.6511 | 65.39 |
>     | $\mathcal{L}^{(\text{cid})}$ | 0.2527 | 0.6532 | 66.85 |
>     | $\mathcal{L}^{(\text{cida})}$ | 0.2859 | 0.7014 | 68.36 |
>
> - Invalidation of fidelity metrics. Another reason for that we prefer NR-IQA is that the fidelity metric can hardly reflect the real visual quality. This phenomenon is also noticed in multiple-step diffusion SR. We quote comparison results of PURE (ICCV2025) [1] to give an example. With the help of multimodal information and larger parameters, the PURE can restore more precisely than OSEDiff. However, the LPIPS and PSNR metrics are far behind OSEDiff.
>
>     | Method | PSNR | LPIPS | ClipIQA |
>     | - | - | - | - |
>     | OSEDiff | 25.15 | 0.2920 |  0.6686 |
>     | PURE | 22.83 | 0.3821 | 0.6817 |
>
>     [1] Perceive, understand and restore: Real-world image super-resolution with autoregressive multimodal generative models.
>
> 2. **About visual quality**. We provide more intuitively better results in supplementary material.  As to mention images, the GenDR restores more vivid details than the existing one-step diffusion model. However, we acknowledge that the improvement over multiple-step methods is limited but exists.
>
> 3. **About inference efficiency**. Yes, the trick is also accessible for other one-step diffusion models and brings a major time reduction. Compared to other one-step diffusion methods like InvSR, GenDR’s speed (77ms) stems from both lighter base model (no extra VAE's encoder or controlnet) and a simplified pipeline (removal of encoder). We summarize the results in the following table.
>
>     | Method | InvSR | OSEDiff-Ori | OSEDiff-Sim | GenDR-Ori | GenDR-Sim |
>     | - | :-: | :-: | :-: | :-: | :-: |
>     | #Params (M, original pipeline) | 1289 | 1775 | - | 1263 | - |
>     | #Params (M, simple pipeline) | 960 | 949 | 949 | 933 | 933 |
>     | #Time (ms) | 115 | 103 | 69 | 92 | 77 |
>
> 4. **About REPA strategy in SR**. DINO loss (via the REPA strategy) is critical but not the key component for SR task. It is complemented by other losses to ensure fidelity. It aligns UNet extracted feature with DINOv2’s representations, preserving high-level semantics (e.g., object shapes and categories), which prevents semantic drift and accelerates training convergence. In the following table (Table 8 in our revised manuscript), we compare the CiDA with/without REPA loss, where using REPA slightly improves PSNR and LPIPS.
>
>     | Method | PSNR | LPIPS | ClipIQA | MUSIQ |
>     | - | - | - | - | - |
>     | w/o REPA | 22.62 | 0.2899 | 0.7020 | 68.30 |
>     | w/ REPA |  23.18 | 0.2859 | 0.7014 | 68.36 |

---

> > ### Comment · Reviewer_Pt8i · 2025-11-22
> > **Response to author rebuttal**
> >
> > Honestly, the author's rebuttal can't sufficiently convince me, but I'm happy to have a further discussion regarding these questions.
> >
> > As for Q1, we have almost focused on improving the perceptual quality in the field of SR. According to my observation, particularly in OSEDiff, while resulting in obviously perceptual improvement, it also produces more artefacts in some cases. Thus, I think it is still necessary to consider the fidelity in model design.
> >
> > As for Q3, it is a useful trick in diffusion-based SR.
> >
> > As for Q4, it seems that the REPA strategy is not effective enough in SR. However, I still think it is a potential research topic, namely, representation learning in SR.
> >
> > By the way, I keep my positive rating.

---

> ### Author Response · Authors · 2025-11-22
>
> Thanks for your immediate and positive feedback on our GenDR. We would further discuss the mentioned question.
>
> Q1: We agree that it is still necessary to consider the fidelity in model design. We also hope to clarify that the stronger fidelity confinement of CiD and 16-channel VAE provides better fidelity than existing methods. However, these advantages are hard to represent on academic benchmarks, where the degradation settings are extremely severe and the evaluation primarily emphasizes regeneration ability and perceptual IQA scores. To alleviate concerns regarding fidelity, we provide extensive comparisons in more practical scenarios, UGC images SR. In **Figure 17** of our latest manuscript, we provide more visual comparison with OSEDiff on short-form UGC images in the wild. Our GenDR executes faithful reconstruction and delivers the strongest restoration of fine-grained details compared with RealESRGAN and OSEDiff.
>
> Q3: Yes, we would additionally remark that these tricks can cooperate with classifiers, where the classifier figures out the class (face scene, CG scenes), and then GenDR can correspondingly utilize varied text embedding (saved locally) to provide differentiated processing.
>
> Q4: We acknowledge that the REPA strategy provides little help for the GenDR in improving restoration quality since we didn't make task-specific modifications. We believe the limited improvement is due to the poor representation of DINO in providing an understanding of image quality. Specifically, the DINO series is trained with images of varied quality, and they add noise as a data augmentation, which makes the learned distribution unable to sort images by quality. However, in the SR task, the understanding of image quality, especially at the regional level, offers more guidance to improve restoration quality. In the future, we will try more effective representation forms from IQA models.

---

> > ### Author Response · Authors · 2025-12-03
> >
> > Regarding concerns about fidelity, beyond visualization comparison on UGC-SR, we provide more results on $\times2$ SR, which focus more on fidelity since they preserve more details. In general, GenDR achieves better fidelity than OSEDiff, leading to approximately 4dB PSNR and 0.1 LPIPS improvement while maintaining NR-IQA scores. The results support the effectiveness of fidelity confinement used in GenDR.
> >
> > | Method | Scale | PSNR | LPIPS | ClipIQA | MUSIQ |
> > | - | - | - | - | - | - |
> > | Real-ESRGAN | $\times$2 | 28.02 | 0.2062  | 0.6014 |  63.54 |
> > | OSEDiff | $\times$2 | 23.29 | 0.2774 |  0.6214 | 67.56 |
> > | GenDR | $\times$2 |  27.18 | 0.1753 | 0.6152 | 66.82 |

---

### Official Review · Reviewer_8VTQ · 2025-11-01

**Soundness:** 3
**Presentation:** 3
**Contribution:** 3
**Rating:** 8
**Confidence:** 5

**Summary:**

The paper introduces GenDR, a framework designed to convert a pre-trained text-to-image diffusion model (such as Stable Diffusion) into a compact model for real-world image super-resolution. The approach uses a 16-channel VAE to retain more detailed image representations and applies CiDA  to align the training objective with the super-resolution task. GenDR produces results in a single inference step and shows improved performance over several baseline methods in terms of perceptual quality and runtime efficiency on benchmark datasets.

**Strengths:**

- The paper addresses a practical limitation of the commonly used f8c4 VAE in SR tasks by expanding it to 16 channels, which helps preserve more details.

- The proposed CiDA framework is a well-engineered solution that enables stable training and efficient inference even in a higher-dimensional latent space.

- The experiments are thorough, covering model size, inference speed, ablations on CiDA, prompt simplification, and performance after SR-specific fine-tuning. These results support both the effectiveness and practical relevance of the method.

**Weaknesses:**

- In Table 2, the paper only reports changes in no-reference metrics, which do reflect perceptual quality improvements, but it lacks reference-based metrics like LPIPS. Including trends in LPIPS (even if degraded) would provide a more complete picture, especially considering the known trade-offs between no-reference and reference-based measures. Visualization comparisons under different loss functions would also strengthen the argument.

- It remains unclear why the model, after being fine-tuned for SR on a 512-resolution UNet, can still generate coherent 1024-resolution T2I results without typical artifacts (e.g., extra limbs). Some clarification on how the SR tuning impacts generalization to higher-resolution T2I tasks would be helpful.

**Questions:**

I suggest the authors include a brief comparison with TVTSR [1] (ICCV 2025) in the main text. While the motivation is aligned, the two methods adopt different and complementary strategies: the proposed approach keeps the spatial compression ratio fixed but increases the channel dimension by 4×, whereas TVTSR maintains the channel size and compresses the spatial resolution by 2×. A short discussion of these orthogonal design choices would help clarify the novelty and positioning of the proposed method.

[1] https://arxiv.org/pdf/2507.20291

---

> ### Author Response · Authors · 2025-11-18
>
> We sincerely appreciate the positive assessment of GenDR’s innovations, efficiency, and experimental advancements. We also acknowledge the valuable feedback, which we address in detail below.
>
> 1. **About the ablation study on CiDA with perceptual quality assessment**. We provide more comprehensive results in **Table 8** of our revised supplementary, including a comparison under perceptual quality. In summary, the perceptual quality (LPIPS) of CiD is better than VSD but slightly behind one training with perceptual loss and MSE loss. Moreover, aligning with our expectation, adding adversarial loss will introduce a performance drop by about 0.03 but also bring huge no-reference improvement (0.05 on ClipIQA and 1.5 on MUSIQ). We provide simplified results here for quick reference.
>
>     | Loss | LPIPS | ClipIQA | MUSIQ |
>     | - | - | - | - |
>     | OSEDiff | 0.3036 | 0.6829 | 67.30 |
>     | $\mathcal{L}^{(\text{mse})} + \mathcal{L}^{(\text{per})}$ | 0.2494 | 0.5235 | 58.19 |
>     | $\mathcal{L}^{(\text{adv})}$ | 0.2731 | 0.6187 | 62.58 |
>     | $\mathcal{L}^{(\text{vsd})}$ | 0.2744 | 0.6496 | 65.01 |
>     | $\mathcal{L}^{(\text{cid})}$ | 0.2527 | 0.6532 | 66.85 |
>     | $\mathcal{L}^{(\text{cida})}$ | 0.2859 | 0.7014 | 68.36 |
>
> 2. **About 1024-resolution generation**. In fact, the 1024px T2I results are generated by the base model (SD2.1-VAE16) rather than GenDR. Notably, SD2.1-VAE16 is trained on both 512×512 and 1024×1024 images. Regarding 1024px SR results, the GenDR is derived from and distilled from the SD2.1-VAE16, which is capable of processing varying resolutions.
>
> 3. **About the comparison with TVTSR**. We have also taken note of this insightful work and incorporated TVTSR into our revised manuscript. Generally, GenDR transfers the latent space for UNet on the T2I task to improve fidelity, while TVT tries to solve the problem from the orthogonal side, training a new VAE (compresses the spatial resolution by 2×). The two approaches have distinct advantages and disadvantages, as detailed below:
>
> - Training VAE  is more accessible since it requires fewer training resources. The removal of VAE's middle block brings additional computation reduction. However, it involves more modifications to a certain pipeline, e.g., additional residual connections and blocks for UNet, which requires more adaptation to be deployed in industry. Moreover, the distillation teacher is restricted to UNet-based diffusion models, e.g., SD2.1, SDXL, showing inferior capability of learning for recent DiT.
>
> - Training UNet (Latent-space transferring) has a higher performance upper bound, better generality, and a more simplified pipeline. Specifically, we can transfer the latent space of SD2.1/SDXL to one of SD3.5 or even Qwen-Image and use DiT as a teacher to conduct CiDA distillation. In our revised manuscript, we provide a primary experiment of pruned-SDXL-based GenDR in A.5, which is founded on SD3.5's latent space and distilled by SD3.5-large with CiD.

---

> > ### Comment · Reviewer_8VTQ · 2025-11-26
> >
> > I have read the authors' rebuttal and the revised manuscript. The authors have effectively addressed my concerns. The inclusion of LPIPS in the supplementary material provides a more comprehensive evaluation, and the explanation regarding the trade-off between different metrics is reasonable. I also appreciate the clarification that the resolution capability is inherited from the base model, which resolves my confusion. Furthermore, the added discussion on TVTSR is insightful; it clearly articulates the advantages of the proposed method regarding deployment flexibility and architectural generality. Given that my concerns are resolved and the paper presents a solid contribution, I will maintain my score of 8.

---

> > > ### Author Response · Authors · 2025-11-26
> > >
> > > Thank you for your thorough and professional review, as well as your unwavering support for our work. Your feedback has been invaluable in guiding us through the revisions, and we deeply appreciate the time and effort invested in assessing our research.

---

### Author Response · Authors · 2025-12-03

Dear Area Chair,

We deeply appreciate your willingness to dedicate your valuable time to conducting a thorough review of our submission and rebuttal amidst the unforeseen circumstances.

To assist your evaluation, we provide a brief summary of our work.

**Paper Overview**:

We introduce GenDR for image super-resolution that focuses on the restoration and preservation of fine details. Our work highlights the key differences between the text-to-image (T2I) and super-resolution (SR) tasks within the diffusion framework. Specifically, we argue that efficient SR diffusion models require a higher latent dimension for effective reconstruction, but a shorter diffusion pathway for efficient generation. Building on this insight, we implement three key modifications to the SD2.1 framework:

1. **SD2.1-VAE16**: We transfer the latent space from 4-channel to16-channel to improve the capablity of preserving details.

2. **CiD/CiDA**: We propose consistent score identity distillation to shorten the diffusion step, a SR-tailored distillation framework that incorporates SR task-specific prior into score distillation and enhances it with adversarial learning and representation alignment.

3. **Simplified pipeline**: We construct a simplified and deployment-friendly pipeline comprised solely of VAE and UNet.

**Initial Ratings**: 8 (8VTQ); 6 (Pt8i); 4 (5Lfg); 6 (kCw1)

**Consensus on Strengths**:

1. Effectiveness and necessity of 16-channel VAE. (All reviewers)

2. Practical and well-engineered distillation framework. (8VTQ, kCw1)

3. Comprehensive experimental evaluation: (8VTQ, 5Lfg, kCw1)

**Main Concerns and Rebuttal Actions**:

1. **Perceptual ablation of distillation strategies (8VTQ)**: We provide more comprehensive results in Tab. 8 to provide a more detailed comparison of varied distillation strategies.

2. **Experimental performance of GenDR (Pt8i)**: We explain that GenDR is subjective-oriented and focuses on visual quality, while it still shows an advance in perceptual quality compared to existing methods. We also include additional qualitative (Fig. 17) and quantitative comparisons to support our explanation.

3. **Method clarity (5Lfg)**: We clarify our method in terms of coherence, relevance, and method density and provide a detailed explanation for the used equation. We also clarify the contribution of CiD to stabilize training and improve consistency via a comprehensive theoretical derivation and discuss its difference with supervised fine-tuning.

4. **Discussion for VAE (kCw1)**. We provide more comparisons for varied VAEs in Fig. 11 and Tab. 5, and we discuss the trade-offs between rFID and gFID.

5. **Inference and training cost (Pt8i, kCw1)**: We provide a detailed quantitative comparison and analysis (Tabs. 9&10) of memory usage and latency for both training and inference.

We sincerely thank all reviewers for their constructive feedback. We believe that our revisions, supplementary experiments, and detailed responses have meticulously addressed these concerns. We kindly request that our rebuttal be fully considered in your final decision, and we are truly grateful for your time and expertise throughout this review process.

Best regards,

Authors

---

### Meta-Review · Area_Chair_j5hB · 2026-01-06

**Summary:**

This paper explores a diffusion-based approach for real-world image super-resolution aimed at improving both detail fidelity and inference efficiency. The method introduces a 16-channel VAE to better preserve image details and applies CiDA to enable stable training and efficient inference. The paper offers a practical and efficient solution for diffusion-based SR and received an initial average rating of 6.0 (4, 6, 6, 8). After the rebuttal, most reviewer concerns were addressed, and no major issues remain at this stage. The final reviewers’ ratings are more positive overall. Given the sound technical contribution and the positive feedback, the AC recommends Accept.

**Reviewer Concerns:**

Reviewers raised concerns regarding performance on reference-based metrics such as LPIPS and PSNR (8VTQ, Pt8i), inferior visual quality compared to prior methods (Pt8i), poor presentation quality and clarity of the methodology (5Lfg), the relevance of CiDA to the SR task (5Lfg), discussion and evaluation of VAE variants (kCw1), as well as inference and training cost (Pt8i, kCw1).

In response to these comments, the authors provided additional experiments, discussions, and results in the rebuttal and revised paper. Most of the concerns listed above were sufficiently addressed. That said, Reviewer 5Lfg continued to express concerns about the paper’s organization and clarity. While the authors made an effort to clarify and revise the presentation during the rebuttal, the reviewer noted that the paper would still benefit from further polishing to make it easier to follow and to better highlight its main contributions.

**Reviewer Scores:**

Reviewers 8VTQ, Pt8i, and kCw1 gave positive initial ratings of 8, 6, and 6, respectively, and did not raise further concerns after the rebuttal, thus their scores would be expected to remain unchanged. Reviewer 5Lfg (initial score: 4) may continued to express concerns about the writing and presentation, and would likely maintain the original score. As a result, the overall rating would remain at 6.0 (4, 6, 6, 8).

---

### Decision · Program_Chairs · 2026-01-26

Accept (Poster)